# Mangrove reforestation provides greater blue carbon benefit than afforestation for mitigating global climate change

Shanshan Song [1,2], Yali Ding [1] ✉, Wei Li [1] ✉, Yuchen Meng [1,2], Jian Zhou[1], Ruikun Gou[1,2], Conghe Zhang[1,2], Shengbin Ye[1,2], Neil Saintilan[3], Ken W. Krauss [4], Stephen Crooks [5], Shuguo Lv[6] & Guanghui Lin [1,2] ✉

Significant efforts have been invested to restore mangrove forests worldwide through reforestation and afforestation. However, blue carbon benefit has not been compared between these two silvicultural pathways at the global scale. Here, we integrated results from direct field measurements of over 370 restoration sites around the world to show that mangrove reforestation (reestablishing mangroves where they previously colonized) had a greater carbon storage potential per hectare than afforestation (establishing mangroves where not previously mangrove). Greater carbon accumulation was mainly attributed to favorable intertidal positioning, higher nitrogen availability, and lower salinity at most reforestation sites. Reforestation of all physically feasible areas in the deforested mangrove regions of the world could promote the uptake of 671.5–688.8 Tg $CO_2$-eq globally over a 40-year period, 60% more than afforesting the same global area on tidal flats (more marginal sites). Along with avoiding conflicts of habitat conversion, mangrove reforestation should be given priority when designing nature-based solutions for mitigating global climate change.

Considering the significant potential for carbon sequestration and greenhouse gas offsets, blue carbon ecosystems, such as mangrove forests, saltmarshes, seagrass beds, and upper estuarine tidal wetlands, have gained global prominence for climate mitigation as a nature-based solution[1]. Since the 1970s, mangrove restoration projects have been initiated in regions such as Southeast Asia, East Asia, and South America[2,3]. These restoration projects were carried out according to local policy and environmental settings[3], whilst they could be generally classified into two categories by silvicultural design: reforestation or afforestation[4]. Reforestation refers to expediently restoring mangroves in areas that suffered from fairly recent degradation or

deforestation by anthropogenic and natural factors[5,6], while afforestation refers to establishing mangroves in areas where mangroves did not previously exist[7,8]. Generally, the trade-offs in choosing between reforestation and afforestation relate to the recovery efficiency and the sociopolitical complexities of land tenure[9,10]. Reforestation may avoid conflicts of converting other vulnerable ecosystems to mangroves but still face tenurial problems, especially for the abandoned maricultural ponds. Afforestation sites (e.g., mudflats) may have lower land costs, but the survival rate of seedlings is often low because of inappropriate hydrodynamic conditions. Quantification of possible differences in ecosystem benefits such as carbon

[1]Department of Earth System Science, Ministry of Education Key Laboratory for Earth System Modeling, Institute for Global Change Studies, Tsinghua University, Beijing 100084, China. [2]Institute of Ocean Engineering, Shenzhen International Graduate School, Tsinghua University, Shenzhen, Guangdong 518055, China. [3]School of Natural Sciences, Macquarie University, Sydney, NSW, Australia. [4]U.S. Geological Survey, Wetland and Aquatic Research Center, 700 Cajundome Blvd., Lafayette, LA 70506, USA. [5]Silvestrum Climate Associates LLC, 1 Crescent Ave, Sausalito, CA 94965, USA. [6]Institute of Marine Ecology and Environment and Hainan International Blue Carbon Research Center, Hainan Academy of Environmental Sciences, Haikou, Hainan 570100, China. ✉e-mail: dingyali@mail.tsinghua.edu.cn; wli2019@tsinghua.edu.cn; lingh@tsinghua.edu.cn

sequestration between mangrove reforestation and afforestation is thus crucial for formulating policies and plans that specify mangrove restoration protocols to enhance blue carbon potential and help with mitigating global climate change.

Mangrove restoration pathways that maximize carbon sequestration potential have been explored, but these have mostly postulated questions related to tree species selection[11,12], or how attaining restoration might occur most efficiently (e.g., natural process vs. active facilitation; monoculture vs. mixed-species stocking; planting density related to carbon and sedimentation)[12–15], with few assessments focusing on the role that establishment location, past land tenure, or silvicultural action may have. While climate factors were found to influence mangrove growth on the continental scale, regional and local factors such as ecogeomorphic settings and environmental conditions are also important[16]. Prior land use certainly influences restoration trajectories in terrestrial forests[17,18]. Whether mangroves previously grew on a site might also reflect the local geomorphic and biophysical property constraints, which would further influence mangrove succession and carbon flux dynamics. For instance, in regions that suffered historical hypersalination-driven death after road levee construction, restored mangroves could store greater amounts of organic matter and nutrients in sediments than currently conserved mangrove area[19]. In addition, varying hydrogeomorphic settings and nutrient availabilities might also influence the carbon stocks of mangrove forests through growth adjustments[20,21]. When restored in aquacultural ponds with high antecedent productivity, mangroves exhibited higher biomass carbon sequestration rates compared with those on less productive sites even though aquaculture previously dominated[6]. Therefore, we hypothesize that trajectories of carbon accumulation in mangroves may be significantly different between reforestation and afforestation actions, given the differences in the suitability of preexisting versus novel settings for mangrove biomass development. However, to the best of our knowledge, no study has addressed this hypothesis at scales necessary to elicit such consideration.

In this analysis, we compiled data on standing stocks of the different ecosystem carbon pools, including aboveground biomass carbon (AGC), belowground biomass carbon (BGC), and sediment carbon (SCS) to a 1 m depth from 379 sites undergoing restoration (Fig. 1). These sites represent 106 scientific publications, cover most of the mangrove regions across the globe[2], and are distributed as follows: Asia Pacific Ocean (69.7%), Asia Indian Ocean (18.5%), Africa Indian Ocean (8.2%), American Atlantic Ocean (2.1%), and American Pacific Ocean (1.6%). Mangrove sites were located between 38 °S and 28 °N, with the mean carbon density of belowground biomass and sediment carbon pools peaking near the equator (0–5 °N) and decreasing toward higher latitudes. For the aboveground biomass carbon pool, mean carbon density was bimodal with a peak around 5 °N and a lower peak near 20°N (Fig. 1). Maximum duration of mangrove afforestation projects spanned nearly 80 years, with the oldest known afforestation chronosequence (of *Rhizophora mangle*) located in Hawaii, USA, where no mangroves have colonized naturally[8]. The majority of stand ages for mangrove reforestation sites were <40 years, owing to silvicultural action to harvest at specific rotation ages. Single mangrove species were used for reforestation (56.7%) and afforestation (80.4%) projects in the majority of cases examined (Supplementary Fig. 1).

## Results and discussion

### Mangrove carbon accumulation after reforestation and afforestation

During the first 40 years of mangrove restoration, aboveground biomass carbon (AGC) in reforestation and afforestation projects both increased gradually (Fig. 2a), but with reforestation sites exhibiting a larger increase in slope than afforestation projects, most prominently after 15 years ($P \le 0.001$, log (age); $P \le 0.001$, log (age) × restoration pathways; Supplementary Table 1). In contrast to AGC, we found no

significant difference between reforestation and afforestation projects for belowground biomass carbon (BGC) increase during the same 40-year period (Fig. 2b and Supplementary Table 1), indicating that similar carbon accumulation rates might prevail among the BGC pool. However, for the top meter of sediment carbon (SCS), carbon density in mangrove reforestation sites was nearly twice that of afforestation sites, in the beginning 5 years after restoration action ($220.7 \pm 38.5$ vs. $108.9 \pm 5.8$ Mg C ha$^{-1}$; mean ± s.e; $P \le 0.01$; Fig. 2c), and this differentiation in sediment properties contributed most to greater realized carbon among reforestation projects. Notably, even by 20 to 40 years, the sediment carbon density at mangrove reforestation sites was more than twice of that at afforestation sites ($293.4 \pm 26.4$ vs. $128.8 \pm 14.4$ Mg C ha$^{-1}$; $P \le 0.001$; Fig. 2c). After excluding the initial sediment carbon storage before restoration, the sediment carbon increments since the time of reforestation are still higher than those for afforestation, although the increments for reforestation tend to be smaller as mangrove growing older (Supplementary Fig. 2).

Thus, the larger sediment carbon pool contributed strongly to the pronounced difference in total ecosystem carbon stocks between reforestation and afforestation projects. Reforestation action supported $232.8 \pm 48.0$ to $407.0 \pm 23.7$ Mg C ha$^{-1}$ and afforestation action supported $119.2 \pm 20.3$ to $213.7 \pm 30.7$ Mg C ha$^{-1}$ over the first 40 years (Fig. 2d; $P \le 0.05$, log (age) × restoration pathways; Supplementary Table 1), largely attributed to the rapid development of aboveground biomass carbon and sediment carbon. Mangrove reforestation is likely a more effective strategy in support of climate change mitigation actions, which may simultaneously avoid further losses of in-situ sediment carbon existing before the reforestation action. It also provides better wood production options when under rotation-age-based logging management[22]. The aboveground biomass carbon accumulation rate first increases with age, peaks around 10–15 years, and then decreases (Supplementary Fig. 3). Therefore, an effective rotation age for maximizing continuous wood production benefit (i.e., maximum wood production per year) might be around 10–15 years, but certainly less than 40 years. However, the logging practices would return some carbon to the atmosphere through residual wood decay and pulsed lateral carbon fluxes, and thus reduce the net carbon benefit in terms of climate mitigation.

### Factors controlling mangrove carbon accumulation

Mangrove carbon accumulation is regulated by climate factors as well as local ecogeographic settings[16]. Thus, aboveground biomass accumulation at both reforestation and afforestation sites can be promoted by an increase in mean annual precipitation (MAP). However, the similar distributions of MAP in the two restoration pathways and their insignificant interactions indicated that MAP is not the main reason for the differences in carbon sequestration between reforestation and afforestation (Fig. 3a, b; Supplementary Note 1, Supplementary Table 2, and Supplementary Fig. 4).

Reforestation is conducted where mangroves previously (and fairly recently) occupied a site before being harvested or destroyed by alternate means. After ecesis, woody debris and antecedent anaerobic conditions are more likely indicative of mangrove development requirements to contribute to more organic matter accumulation and less nitrogen degradation of the sediment, thus supporting the potential for greater productivity on reforestation sites[19]. Our analysis supports this net influence. Further, the initial sediment characteristics of reforestation sites indeed had greater total organic carbon (TOC) and total nitrogen content (TN) compared with afforested project sites (Supplementary Fig. 5). Furthermore, during successional growth periods, sediment in the reforestation sites continued to exhibit greater amounts of TOC and TN ($P \le 0.001$, Fig. 3). Greater sediment fertility would promote mangrove shoot growth, canopy leaf area expansion, and its associated carbon storage potential[23], as indicated by the significant positive relationship between AGC and TN ($P \le 0.05$, Fig. 3).

Mangrove ecosystem reforestation projects were mainly located among higher intertidal zones that were accessible to local residents to deforest, while tidal flat or seagrasses favored for afforestation were mainly located within lower intertidal zones[24]. Lower sediment carbon-to-nitrogen ratio (C:N) of afforestation sites, as we also demonstrated, likely further verified the differentiation in site index between restoration project types and may also be associated with greater potential in-welling of marine-sourced carbonate material or marine phytoplankton to the lower intertidal zone of afforestation sites (Supplementary Fig. 5)[25]. Along with the inherent procedural disparity between silvicultural actions associated with reforestation versus afforestation, this observation further contributes to a rapidly developing understanding that mangroves are far more likely to survive mid-to-upper intertidal positions associated with restoration action than lower intertidal positions flooded >~60% of the year[26]. However, the comparisons of intertidal positions should be interpreted with caution because some reforested ponds might be dug out to have a lower surface elevation[27].

Increasing tidal inundation and transport energy could lead to relatively higher sediment porewater salinity among afforestation projects locally (Fig. 3). During early mangrove growth stages, high salinity often results in reduced stomatal conductance and photosynthetic efficiency, consequently inhibiting seedling growth and rapidity of mangrove ecosystem development if salinity is beyond specific thresholds that also vary by species[28]. For example, *Sonneratia apetala*, a common mangrove species used for reforestation and afforestation projects in Asia, experienced the slowest growth at the highest salinity tested[29]. The mangrove literature is abundant with similar salinity versus growth assessment; therefore, salinity as a constraint to carbon accumulation at a larger scale between reforestation and afforestation projects may warrant further study. Higher salinity stress might lead to greater biomass partitioning to belowground in afforestation sites versus reforestation sites, which might in turn explain the similar rate of belowground productivity among these two project types (Supplementary Note 2 and Supplementary Fig. 6).

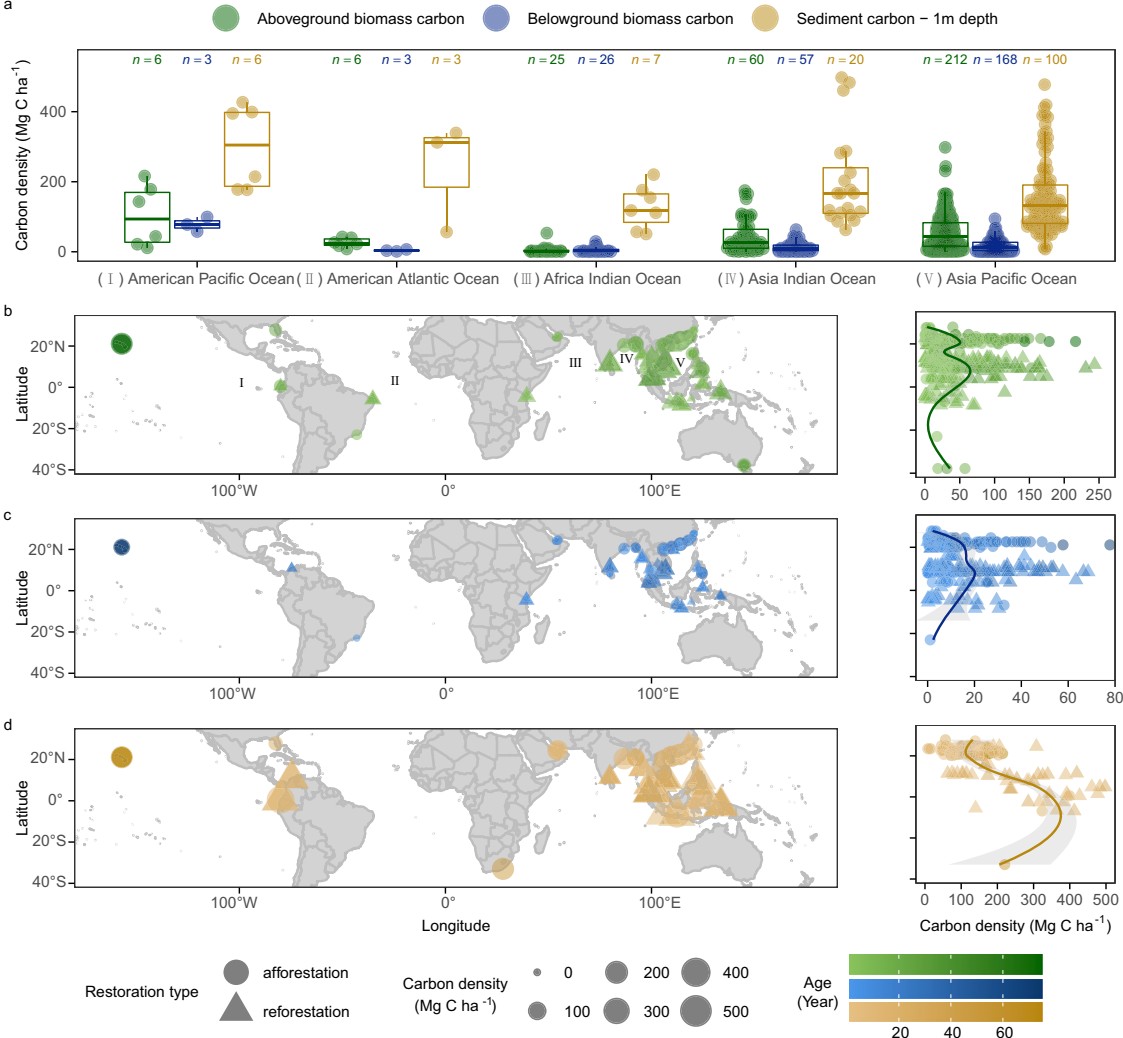

**Fig. 1 | Carbon density among global mangrove ecosystem restoration projects.** Colors represent different carbon pools (green: aboveground biomass carbon; blue: belowground biomass carbon; brown: sediment carbon to 1 m depth). **a** Carbon density distribution of the three carbon pools for mangrove ecosystem restoration projects by geography. For each box plot, individual data points are shown as circles. The center line and the top and bottom of the box represent the median and the interquartile range (25th percentile and 75th percentile). The whiskers represent the minimum and maximum limits. The sample size (*n*) of each group is shown at the top of each box plot. **b**–**d** Spatial distribution of carbon density within each of the three carbon pools. The density of shading represents the mangrove ecosystem restoration project age, with symbol size representing the relative magnitude of carbon density and symbol shape representing the type of mangrove ecosystem restoration project (reforestation or afforestation). Latitudinal trends are presented to the right of **b**–**d**, with shading representing the 95% confidence interval of polynomial regression fits.

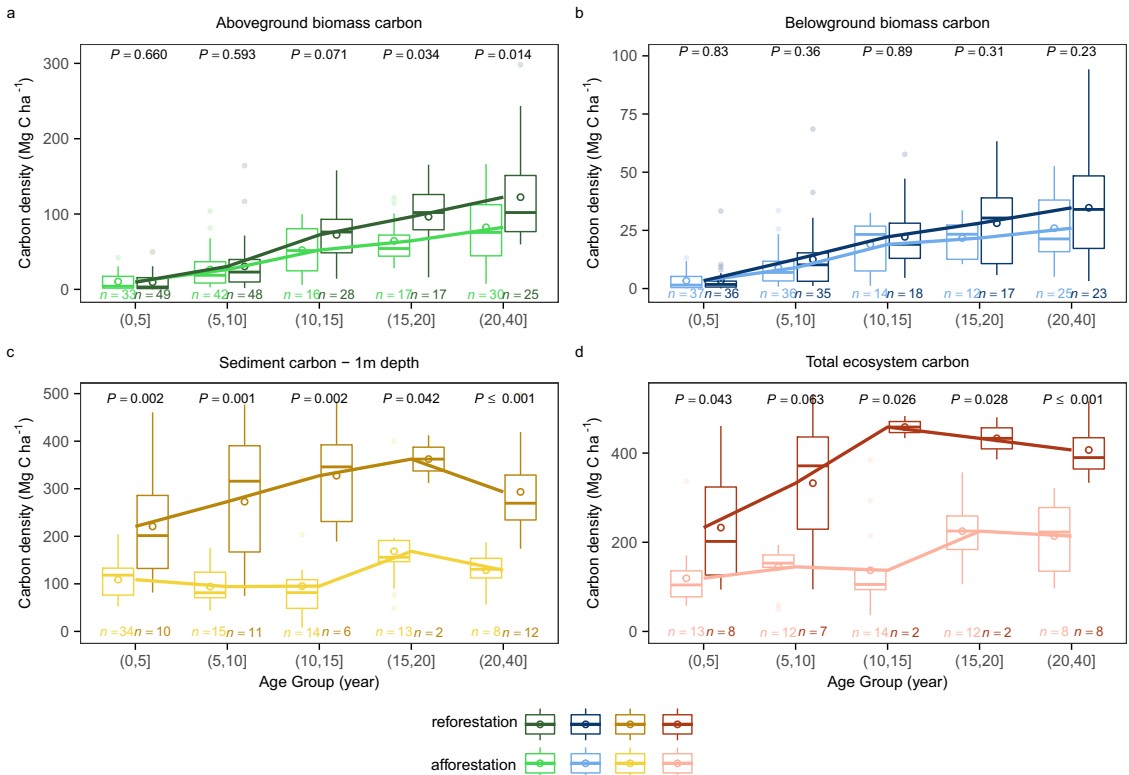

**Fig. 2 | Comparison of carbon density between mangrove reforestation and afforestation projects over a 40-year period. a** Aboveground biomass carbon, **b** Belowground biomass carbon, **c** Sediment carbon to a depth of 1 m, **d** Total ecosystem carbon. The x-axis represents different age groups (0–5, 5–10, 10–15, 15–20, and 20–40 years). For each box plot, components are as detailed in Fig. 1. Outliers are represented by dots. The mean carbon density in each time group is represented by a circle and connected by a line. The sample size (*n*) of each group is shown at the bottom of each box plot. Dark- and light-colored groupings, of any color, represent mangrove reforestation and afforestation, respectively. A significance level of difference between reforestation and afforestation within each age group is calculated by the Wilcoxon two-sided test.

Approximately 42% of carbon burial in the mangrove sediment (SCS) is derived from plant litter[30]. More rapid accumulation of aboveground biomass contributes more leaf litter to the sediment in reforestation sites, which, as we found, are flushed less by the tide to reduce leaf litter export as particulate organic matter than are afforestation sites occurring lower in the intertidal (Fig. 2 and Supplementary Table 1). Moreover, sediment carbon accumulation depends on the balance between carbon input and decomposition[30]. Litter decomposition is largely dependent on tissue quality. Compared with non-succulent leaves, succulent leaves usually store more nitrogen content[31], which is likely to enhance higher microbial decomposition. However, mangrove leaves also have a high concentration of tannins to deter crabs from leaf-herbivory[32]. Despite the succulence characteristic of *Rhizophora* spp. leaves (species dominating reforestation projects globally), its higher tannin and lignocellulose content may hamper litter decomposition, especially when compared with *Kandelia* spp. and *Sonneratia* spp. (species widely used in the afforestation projects, at least in the Indo-Pacific region) with higher content of fatty acids and amino acids[25,33,34], as observed by some litter bag incubation studies[33,35]. Thus, the leaf biochemistry of mangrove species selected for reforestation may promote greater organic matter accumulation in sediments than that for afforestation (Supplementary Fig. 1)[25,36].

**Carbon mitigation potential of global mangrove restoration**

Previous studies have estimated the carbon sequestration rate with succession among mangrove ecosystem restoration sites on a global scale, without considering all ecosystem components or the restoration project type that would contribute to carbon mitigation potential[15,37]. When combining all carbon pools (AGC, BGC, and SCS), our results predicted that reforestation would sequester 60% more carbon per hectare than afforestation over the first 40 years. For a 40-year-old mangrove forest undergoing reforestation, 127.7 Mg C ha$^{-1}$ (110.7–144.7, 95% confidence interval) would be stored in aboveground biomass, while this value would only reach 88.7 Mg C ha$^{-1}$ (70.2–107.2) with afforestation (Fig. 4a). Belowground biomass would not differentiate; i.e., 38.7 Mg C ha$^{-1}$ (30.1–47.3) for reforestation vs. 37.0 Mg C ha$^{-1}$ (22.9–51.2) for afforestation (Fig. 4b). However, for the sediment carbon pool, mangrove reforestation project type promotes carbon storage (or preservation) at an approximate double rate versus afforestation, or 139.2 Mg C ha$^{-1}$ (136.4–142.1) vs. 65.0 Mg C ha$^{-1}$ (−3.7–133.7) (Fig. 4c), assuming adequate hydrologic function to maintain mangrove development after reforestation. By way of annual increment, this difference would equate to 3.5 vs. 1.6 Mg C ha$^{-1}$ yr$^{-1}$ of sediment carbon accumulation for reforestation and afforestation sites, respectively, which is in the range measured by radiometric geo-chronologies and surface elevation table methods (0.06–17.2 Mg C ha$^{-1}$ yr$^{-1}$) among naturally occurring mangrove forests[37].

Globally, mangrove deforestation area was estimated as 904,953 ha for the period of 1996 to 2016[38], which was reduced to 614,467 ha as a biophysical constraint when eroded and developed locations where mangrove restoration has no potential are excluded[39,40]. Assuming all these areas to be reforested within 1 year, the cumulative carbon sequestration over the next 40 years is 688.8 (624.8–752.7) Tg CO$_2$-eq, which is about 259 Tg CO$_2$-eq higher than afforestation (Figs. 4e, 1-year-completed restoration scenario, see Methods). Indonesia, Mexico, and Myanmar will provide the greatest carbon mitigation potential through reforestation (188.7, 110.4, and 61.3 Tg CO$_2$-eq over 40 years, respectively) (Fig. 4e). Under other scenarios with slower rates of restoration area, i.e., reforesting all these areas within a 5-year and 10-year period, the cumulative carbon

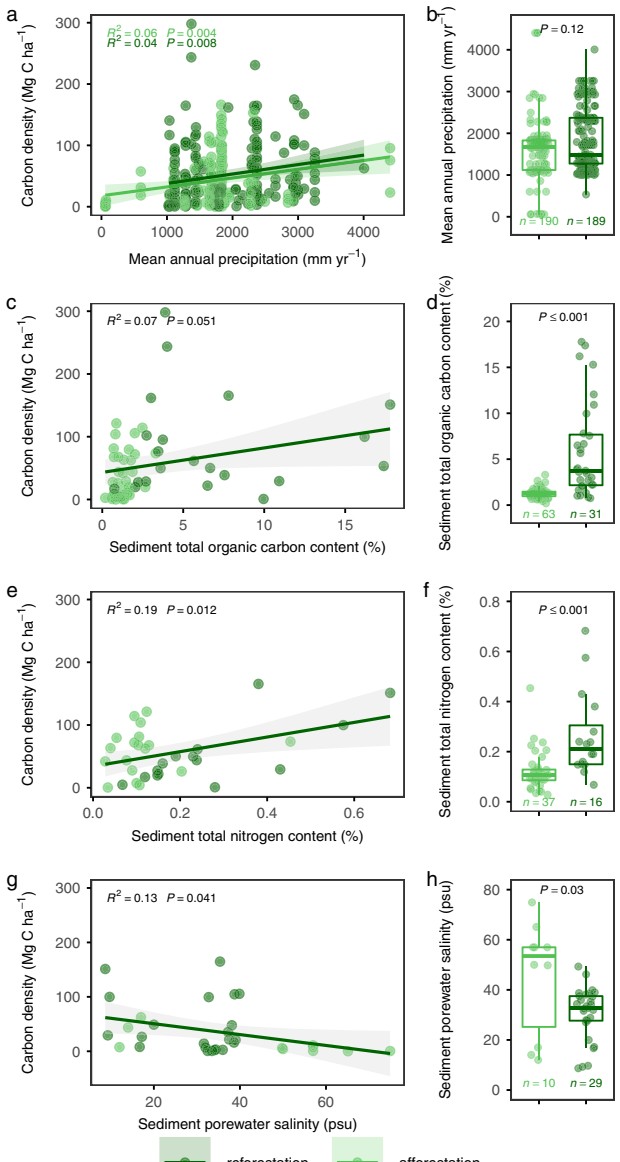

**Fig. 3 | Relationship of aboveground biomass carbon density with mean annual precipitation and sediment properties for mangrove restoration.**
**a**, **c**, **e**, **g**, Relationship of aboveground biomass carbon (AGC) density with mean annual precipitation (MAP), sediment total organic carbon content, total nitrogen content, and porewater salinity. Lines in **a**, **c**, **e**, **g** represent the ordinary least squares regression, with shading representing the 95% confidence interval of regression fits. psu refers to practical salinity units. **b**, **d**, **f**, **h**, Comparisons of MAP and sediment properties between reforestation and afforestation sites. Note that some sites with belowground biomass or sediment organic carbon density but without aboveground biomass data are also included in **b**, **d**, **f**, **h**. For each box plot, components are as detailed in Fig. 1. Significance level of difference between reforestation and afforestation is calculated by the Wilcoxon two-sided test.

sequestration potential during 2021–2060 is 687.2 (627.3–747.1) and 684.2 (631.5–736.9) Tg $CO_2$-eq, respectively, and it reaches 671.5 (637.1–706.0) Tg $CO_2$-eq in the scenario with varying rates of restoration area across countries. Our estimate is lower than a previous study (1338 Tg $CO_2$-eq)[41], because they used biomass of mature mangroves as the biomass sequestration potential and also included the avoided soil carbon loss (instead of the new soil carbon accumulation since restoration). In terms of annual ecosystem productivity, restoring all potentially available mangrove areas (reforestation) would increase annual $CO_2$ uptake of existing estuarine and coastal wetland

ecosystems by 4.3–5.1%[42], highlighting the potential blue carbon sink strength of mangrove reforestation, especially in regions suffering high levels of recent deforestation.

Additionally, successful afforestation actions often convert habitats with inherent value (mudflats, seagrasses) to another habitat—herein, mangroves—with the previous habitat types having its own but not overlapping cadre of values[10,43]. The necessity of converting one valuable habitat into another has been questioned for decades[3,43]. Intertidal mudflats themselves could provide multiple ecosystem functions not provided by mangroves, such as repositories for juvenile fishes and refueling sites for shorebirds and threatened migratory waterbirds[44,45]. Furthermore, it is often more cost-effective to rehabilitate or reforest existing mangrove areas than to convert or create new mangrove habitats when only considering implementation cost[46]. Therefore, avoiding the potential for habitat conversion would also give prioritization to reforestation over afforestation.

## Limitations and potential caveats
While our analysis included data from over 370 restoration sites globally, there was a more limited number of observations from key regions, such as Africa, Australia, and North America (Fig. 1), where mangroves also suffered from substantial deforestation[40] or are expanding into saltmarsh[47]. The omission was partly because some accounts from available mangrove restoration sites were missing key characteristics (e.g., age, land use history, Supplementary Fig. 7). Therefore, data that may expand spatial distribution, temporal duration, and parameter integrity would reduce uncertainty for global inference and provide better guidance. Mangroves can be quite old, and while a 40-year chronosequence would likely represent many silvicultural rotation ages, this time period would not properly canvass restoration carbon dynamics associated with long-term ecosystem restoration goals as would be specified when opting for restoration or afforestation as a nature-based solution to reduce, e.g., erosive wave energy.

Reforestation sites in our analysis were also more concentrated in tropical zones, while afforestation sites were scattered throughout both tropical and subtropical zones, leading to climate as a possible covariate. To test the potential climate effect, we further analyzed the influence of MAP and mean annual temperature (MAT) on mangrove carbon accumulation, in addition to the restoration pathways (Supplementary Note 1). The results show that aboveground biomass carbon accumulation during mangrove reforestation and afforestation is influenced by MAP to a similar extent (see details in Supplementary Note 1). In contrast to aboveground biomass, belowground biomass carbon is influenced by MAP but not by restoration pathways (Supplementary Table 2 and Supplementary Fig. 4), consistent with the similar belowground biomass carbon accumulation between the two pathways (Fig. 2).

What our analysis clarified was that the mean carbon sink strength for both silvicultural pathways varied; within-pathway variation would also exist as managers opt to use a wide variety of silvicultural practices around the world. Neglecting the within-pathway variation is also likely to misestimate the carbon sink effect in some cases. For example, in an estuarine tidal flat actively undergoing sedimentation, *Kandelia* spp. afforested for 3–6 years exhibited similar growth characteristics as a nearby reforested site in abandoned maricultural ponds (Supplementary Fig. 8), probably because of relatively greater riverine freshwater inputs and salinity relief of the afforested tidal flat. Additionally, sediment textures of abandoned ponds usually depend on the local hydrogeomorphic conditions associated with their original mangrove setting[6]. For example, the substrate of abandoned fish ponds is mainly gravelly sand in sandstone coastal regions[48], but mainly clay-loam or silt clay-loam in riverine-influenced areas[49]. These variations of edaphic and hydrological properties could lead to plastic but species-specific dynamics of the above- and belowground biomass[27,50]. Therefore, more detailed information on classifications within each restoration

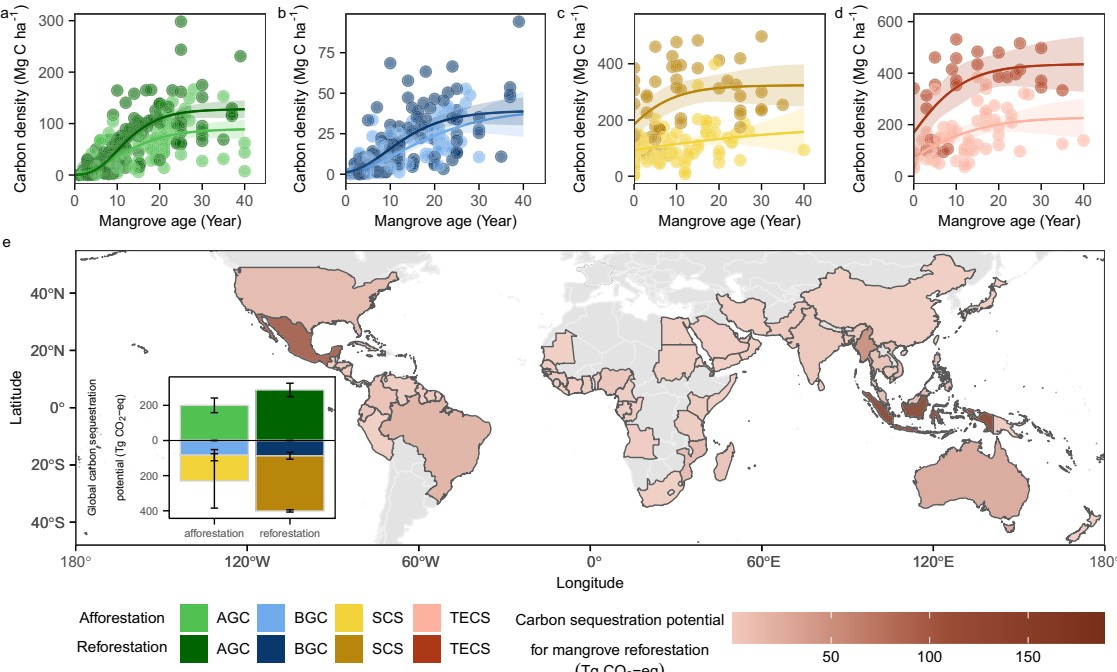

**Fig. 4 | Carbon sequestration potential if mangrove afforestation and reforestation projects are implemented globally over a 40-year period. a–d** Changes in carbon density among pools over time during mangrove reforestation and afforestation. **a** Aboveground biomass carbon (AGC) pool (*n* = 167 for reforestation, *n* = 138 for afforestation); **b** Belowground biomass carbon (BGC) pool (*n* = 129 for reforestation, *n* = 124 for afforestation); **c** Sediment carbon (SCS) pool (*n* = 50 for reforestation, *n* = 108 for afforestation); and **d** Total ecosystem carbon (TECS) pool (*n* = 28 for reforestation, *n* = 63 for afforestation). Fitted lines and ribbons are calculated as the predicted values and the 95% confidence intervals for the 2.5th and 97.5th percentiles based on nonlinear models. **e** Country-specific maximum carbon mitigation potential from mangrove reforestation over 40 years (2021–2060) under the 1-year-completed restoration scenario, with the density of shading representing the maximum carbon mitigation potential of each country. Countries in gray had no mangrove loss to deforestation between 1996 and 2016, or no mangroves. The inset graph represents the global maximum carbon mitigation potential by reforestation and assuming the same area for redeeming afforestation. Error bars represent a 95% confidence interval for maximum carbon mitigation potential in each carbon pool.

pathway and on sediment edaphic properties at the site scale can be expected to inform a more accurate selection of restoration sites.

Moreover, we acknowledge that considering the biophysical limitations of mangrove restoration is only the first step to assessing the feasible region for future restoration projects. In this study, we mainly focus on the benefits of carbon storage from mangrove restoration, but there are also many other factors such as social policies and economic costs that can constrain the implementation of mangrove restoration[41,51]. In particular, some recently deforested mangrove wetlands experience high pressure from land use change, and the economic and social benefits provided by land use after deforestation would raise the opportunity costs of mangrove reforestation[52]. For example, the potential payments for the economic loss of agriculture and pasture production when converting deforested lands to mangrove wetlands would reach US$ 1,687–47,589 per ha in a 40-year period[52]. Unless with support from government policies and economic compensation for mangrove restoration, the high opportunity cost would eliminate the desire of local communities and landowners to engage in mangrove reforestation activities in the deforested regions[51]. On the other hand, even though some deforested lands have been abandoned, it is still uncertain whether it can be reforested due to the debate over land tenure for individual projects[9]. Under the constraints of these factors, therefore, the achievable carbon sequestration potential of global mangrove reforestation might be lower than our estimates. However, it is difficult to make an accurate estimate at the global scale, due to the varying local policies and the lack of high-resolution land value data. Finally, we only considered regions with mangrove deforestation after 1996 as the possible land source for reforestation due to the data limitation, but high rates of global mangrove area loss in the

1980s and 1990s[53,54] were reported, which could provide potentially a more available area for mangrove reforestation.

It is notable that global mangrove area losses have slowed over the last decade or two, and some countries like China have even realized a net increase in mangrove area. Restoration has a prominent role in area-offsets contributing to net reductions in global mangrove losses. By comparing the carbon accumulation trajectories of different mangrove silvicultural pathways, we demonstrate in this global analysis that reforesting mangroves on previously degraded or converted sites provides greater benefit to carbon sequestration than afforesting tidal flats or other marginal locations. As nature-based solutions are established to simultaneously confer enhanced blue carbon sink strength for climate change mitigation and minimize disturbance to other ecosystems, disentangling the silvicultural approach, or procedural options for restoration, could benefit the efficiency of future actions.

## Methods
### Literature search and screening
Our analysis included a systematic literature search and was conducted by following the PRISMA protocol[55] (Supplementary Fig. 7). We searched through Web of Science and China National Knowledge Infrastructure (CNKI) platforms by using keywords listed in Supplementary Table 3. A total of 3299 potentially relevant articles were found (Mandarin and English). The availability of peer-reviewed datasets associated with these published articles[11,15,56–59] and online databases (The Sustainable Wetlands Adaptation and Mitigation Program (SWAMP) database, https://www2.cifor.org/swamp) were also considered. We then removed a significant number of articles through title screening, leaving 551 articles for further inspection.

For these remaining articles, we used a four-step critique process to screen their title, abstract, and full text. We determined that firstly, they must provide carbon density data for at least one of the four mangrove carbon pools (i.e., aboveground biomass, belowground biomass, sediment organic carbon, or total ecosystem carbon). Secondly, articles needed to state the forest age or the starting date of the restoration action. For those studies providing only age intervals (e.g., 10–25 years, >66 years), we excluded them from the analysis. Thirdly, a description of prior land use was required. From these, mangrove restoration could be divided into two categories—reforestation and afforestation—on whether mangroves previously existed in that location. For reforestation, the initial conditions for inclusion were: (1) abandoned agricultural/aquacultural sites built previously by excavating mangrove forests, (2) clear-felled mangrove lands after wars, timber harvest, and silvicultural management, and (3) mangrove forests with mortality due to spraying of defoliants and hydrological alteration caused by the construction of embankments. We compared the carbon densities of reforested mangroves among sites with different causes of degradation/deforestation, and no significant difference is found (Supplementary Fig. 9). For those reforested mangroves, we assumed they would be protected and conserved by local governments and non-government organizations, so that there will not be human-driven degradation or deforestation in the near future. However, we acknowledge that a fraction of mangrove reforestation is managed for wood production, which means logging would happen at a certain interval after reforestation at these sites. For these logging sites, we used their reported measurements after clear-cut, such as 0-, 5-, 10-, 15-, and 25-year post-harvest sites in Sundarbans, Bangladesh[60]. On the other hand, the future occurrence of natural-driven deforestation (e.g., cyclones) is difficult to predict, and thus not considered in our study. For afforestation, the initial condition for inclusion was the presence of non-mangrove habitat immediately before afforestation began, such as mudflats, seagrass, saltmarsh, coral reef, or denuded areas. In most cases, reforestation and afforestation were undertaken through active planting without much re-engineering[4], but for reforestation, natural regeneration could have, and in many places likely did, augment recruitment[61]. Moreover, we only considered mangrove succession that started from near-barren land with an insignificant amount of biomass, and introductions of exotic species to degraded areas with sparse trees were not incorporated. Lastly, if the forest age or prior land use type was not given, the articles needed to specify the location of sampling plots (latitude, longitude). With the coordinates matching, prior land use type and establishment dates were sometimes identifiable through remote sensing (Supplementary Fig. 10). For those articles sharing the same restoration sites but showing different aspects of the data collection, we combined the results and considered the collective work as one source. Based on the space-for-time method, data in the control sites before mangrove restoration actions were also collected as a paired site of restoration (e.g., abandoned ponds before mangrove reforestation; mudflats before mangrove afforestation). In total, we obtained data from 379 mangrove restoration sites described by 106 articles.

## Data extraction

We extracted aboveground living biomass carbon (AGC), belowground living biomass carbon (BGC), sediment carbon (SCS), and total ecosystem carbon (TECS) density from the 106 original data sources. In most cases, numeric values were provided. For those data not provided numerically but graphed, we determined values from figures with the application of GetData Graph Digitizer (http://getdata-graph-digitizer.com/).

Among the articles, aboveground and belowground biomass (Mg ha$^{-1}$) data were obtained using either a harvesting method (empirical) or an allometric method (calculation). Aboveground biomass represented the sum of stem, leaf, and branch dry weight,

and we included prop root biomass when *Rhizophora* spp. were present. For soil coring methods that determined belowground biomass or sediment carbon density, belowground biomass was considered the dry weight of living coarse and fine roots multiplied by the ratio of core area to land surface area[62]. For allometric methods, trunk diameter at breast height (DBH, ~1.3 m) and tree height were used to calculate aboveground and belowground biomass by species-specific or common allometric equations[63]. These equations were also used to calculate the belowground biomass when articles provided plot information (DBH, height) but not belowground biomass (Supplementary Table 4). Total biomass was calculated as the sum of aboveground and belowground biomass. Deadwood and pneumatophore biomass were not included in our analysis; these data are rarely provided and/or methods of determination are inconsistent among global studies[64]. Some articles provided total biomass and shoot/root biomass ratio (S/R), and in such cases, above- and belowground biomass data were obtained through calculation as follows:

$$\text{Aboveground biomass} = \text{Total biomass} \times \frac{\frac{S}{R}}{\frac{S}{R}+1} \tag{1}$$

$$\text{Belowground biomass} = \text{Total biomass} \times \frac{1}{\frac{S}{R}+1} \tag{2}$$

For those articles measuring carbon content, study-specific carbon conversion factors were used to transform biomass to biomass carbon density (Mg C ha$^{-1}$). If carbon content data were not provided, we converted aboveground and belowground biomass to carbon density by applying a conversion of 0.47 and 0.39, respectively[65]. The aboveground biomass carbon density was divided by its corresponding age to get the average aboveground biomass carbon accumulation rate (Mg C ha$^{-1}$ yr$^{-1}$).

For sediment carbon density (SCS, Mg C ha$^{-1}$), we selected the top 1 m because this depth equated to the most commonly reported depth[66], which is also consistent with recent blue carbon standing stock assessment guidance[64,67]. Sediment carbon stock was calculated by multiplying sediment organic carbon content (SOC, %) by bulk density (BD, g cm$^{-3}$), integrated over depth (cm). For studies that reported sediment carbon stock to <1 m depth, we assumed that its organic sediment layer was deeper than 1 m and that the carbon density of the unmeasured depth was the same as that of the deepest measured layer. For example, when the deepest measured sediment layer was 50–60 cm, the carbon density in the 60–100 cm layer was assumed to be the same as that in 50–60 cm, which might overestimate sediment carbon density slightly. Plus, we did not include studies that only measured surface sediment carbon (<20 cm) in our dataset. All biomass carbon data were summed to estimate TECS.

Climate factors (i.e., mean annual temperature, MAT, and mean annual precipitation, MAP) were extracted from the WorldClim2.0 dataset[68] (spatial resolution: 30 s, https://www.worldclim.org/data/worldclim21.html) using the longitude and latitude of each restoration site, and averaged within a 1 km buffer of each restoration site.

## Influence of restoration pathways and climate factors on carbon accumulation

To examine the influence of restoration type on the different mangrove carbon density pools over time, a linear mixed model was used with restoration duration (age) and restoration type (reforestation, afforestation), and their interaction was denoted as a fixed factor as well as their restoration region as a random factor using the lme4 (version 1.1.29) and emmeans (version 1.7.5) packages of R (version 4.0.4, http://www.r-project.org/). Potential nonlinear growth patterns

were modeled with a log transformation of age in comparison to no transformation using Akaike Information Criterion (AIC). Models with lower AIC showed better goodness of fit. The AIC results indicated that log transformation worked better to tease out carbon pool dynamics (Supplementary Tables 1, 2). Analysis of variance (ANOVA) was used to evaluate the significance of each variable at a significant level of 0.05.

Moreover, to assess the difference in recovery trajectories with more detail, we compared the carbon density between mangrove reforestation and afforestation at each stage (age) of mangrove stand development. Ages of sampling plots were divided into 5-year age classes, but were grouped for 20–40 years due to the limited sample size. In contrast to the biomass carbon pool, the sediment carbon pool has various sizes before restoration. To eliminate the influence of initial sediment carbon density, sediment carbon density increments after restoration were also used to compare the carbon accumulation trajectories between these two restoration pathways. Using the space-for-time approach, the difference in sediment carbon densities between the restoration site and the paired control site (without restoration) was considered as the carbon change induced by mangrove restoration. We acknowledge that some uncertainties exist in the space-for-time method because it is difficult to find a perfect control site, which may partly explain the varying, even negative, carbon stock increments in some age groups (Supplementary Fig. 2). To minimize the uncertainties, we corrected the negative values if the carbon accumulation rate was measured simultaneously by $^{210}$Pb[60] and recalculated the carbon density increments via multiplying the carbon accumulation rate by its restoration duration. These carbon density increments were then compared between mangrove reforestation and afforestation among age classes.

Before comparison, normality and homogeneity of model residuals were tested using the Shapiro–Wilk and Levene tests. As assumptions were not met in some groups, the significance of differences was assessed with non-parametric tests. The Wilcoxon two-sided tests were used to compare the difference in carbon density or its increments (mean and standard error) between reforestation and afforestation action for each age group. Comparisons of climate factors and sediment properties were analyzed similarly to carbon density data using the Wilcoxon test. The Kruskal–Wallis test combined with the Bonferroni adjusted post hoc Dunn test was used to compare the differences of more than two groups (i.e., aboveground biomass carbon accumulation rates among the five age groups during mangrove reforestation; carbon densities among mangroves at sites with different causes of degradation/deforestation). All the differences were considered to be significant at a level of $P \leq 0.05$. Finally, relationships among climate factors, sediment properties, and AGC as well as BGC/AGC were determined through the use of ordinary least squares (OLS) regression.

## Plant and sediment carbon accumulation model

The plant growth rate is hypothesized to decrease with stand age and reach an equilibrium (or steady-state) during the latter stages of stand development[22,69]. Therefore, plant biomass would exhibit a nonlinear increase in many cases. Thus, to determine the recovery trajectories of mangrove biomass carbon pools, we fit three nonlinear growth models (i.e., Von Bertalanffy model, Gompertz growth model, and Chapman–Richards model)[70] as follows:

$$\text{Von Bertalanffy model}: C = \text{Asym} \times (1 - e^{-b \times (\text{Age} - c)}) \qquad (3)$$

$$\text{Gompertz growth model}: C = \text{Asym} \times e^{-b \times c^{\text{Age}}} \qquad (4)$$

$$\text{Chapman} - \text{Richards model}: C = \text{Asym} \times (1 - e^{-b \times \text{Age}})^{c} \qquad (5)$$

Where Asym defines the maximum carbon density that a mangrove forest could theoretically reach along an age chronosequence, $b$ and $c$ determine the position and shape of the curve before reaching an asymptote.

When mimicking the sediment carbon pool and total ecosystem carbon pool, carbon density at initial condition was introduced in the growth model as a non-zero baseline value for year 0. Therefore, Chapman–Richards model was unsuitable for these two carbon pools as it defines zero carbon density at the starting year. Combining the model performances in all of our four carbon pools, we used the Gompertz growth model to mimic the carbon accumulation trajectories on both mangrove reforestation and afforestation (Supplementary Table 5 and Supplementary Fig. 11).

All model fitting, comparison, confidence interval calculations, bootstrapping, and integration were conducted using nlme (version 3.1.157), nlstools (version 2.0.0), car (version 3.0.13), stats (version 4.2.0), FSA (version 0.9.3) and rcompanion (version 2.4.18) packages, and visualization procedures were determined with ggplot2 (version 3.3.6) packages in R.

## Global mangrove carbon sequestration potential from restoration action

Since mangrove reforestation action occurs where a mangrove community previously existed, we assumed that any mangrove area loss since 1996 provided same-area reforestation viability. We used the Global Mangrove Watch dataset to define the mangrove deforestation area between 1996 and 2016[38]. Mangrove deforestation was mainly derived from coastal erosion, transformation into settlements, commodities production (e.g., aquacultural/agricultural plots), mangrove clear-cutting operations, or mortality from extreme climate events[40]. The proportion of each deforestation caused by a country and its corresponding exclusive economic zone (EEZ)[71] was calculated as the average estimate between 2000 and 2015 by ref. [40]. For those EEZs that had detectable mangrove loss but were not associated with an associated driver of loss, we used the average proportion of loss drivers in the corresponding geographical zone (Supplementary Table 6). For geographical zones lacking corresponding loss-driver data (i.e., East Asia including China and Japan), we used the average proportion of mangrove loss drivers in representative areas of China to represent East Asia[72,73]. The overall feasible reforestation extent was then calculated as the sum of each kind of deforestation area in each EEZ. As a biophysical constraint, regions experiencing coastal erosion and settlement development were excluded from our analysis because no restoration possibility exists for those areas[39].

The $CO_2$-eq sequestration potential (Seq, Mg $CO_2$-eq ha$^{-1}$) under mangrove reforestation was calculated as the sum of carbon density increments of AGC, BGC, and SCS from the initial baseline (when age was 0), which are predicted by their corresponding growth models and confidence intervals. We used a period of 40 years (i.e., up to 2060) to assess carbon sequestration potential.

On the other hand, four scenarios with different recovery rates were used: (1) 1-year-completed restoration scenario: restoring all these deforested regions as quickly as possible so that all the restored mangroves could fix carbon for 40 years (Eq. (7)); (2) 5-year-averaged restoration scenario: following some short-term targets that countries have pledged, like Indonesia (rehabilitate all of the damaged mangroves (about 600,000 ha) during 2020–2024)[74] and China (restoring 18,800 ha during 2020–2025)[75]. All of these mangrove area losses globally would be restored within 5 years. Restoration effort and project implementation rate in each country was assumed to be the same; therefore, mangrove restored in the first year could fix carbon for 40 years while those restored in the fifth year could fix carbon for 36 years (Eq. (8)); (3) 10-year-averaged restoration scenario: following median-term goals promoted by international forums and organizations, like COP 26 (halt and reverse forest loss and land degradation by

2030, https://ukcop26.org/) and Global Mangrove Alliance (increase global mangrove area by 20% by 2030, https://www.mangrovealliance.org/). All of these mangrove area losses globally would be restored within 10 years. The assumption used for restoring areas each year by country was similar to 5-year-averaged restoration scenario (Eq. (9)); (4) varying rates across countries scenario: assuming countries pledging or agreeing to support mangrove replanting projects on COP 26 would restore their deforested mangrove area within 10 years (i.e., by 2030), except for countries like Indonesia and China assumed to finish in 5 years with their officially-promulgated policies[74,75]. Other countries were assumed to reforest their harvested or damaged mangroves within 20 years (Eq. (10)). We multiplied the $CO_2$-eq sequestration potential in certain time intervals by viable restoration area to indicate the maximum climate change-mitigative carbon storage benefit under each mangrove restoration scenario and pathway.

$$f_{TECS}(i) = f_{AGC}(i) + f_{BGC}(i) + f_{SCS}(i) \qquad (6)$$

$$Seq = Area_{2016-1996} \times f_{TECS}(40) \times \frac{44}{12} \qquad (7)$$

$$Seq = \sum_{i=36}^{40} \frac{Area_{2016-1996}}{5} \times f_{TECS}(i) \times \frac{44}{12} \qquad (8)$$

$$Seq = \sum_{i=31}^{40} \frac{Area_{2016-1996}}{10} \times f_{TECS}(i) \times \frac{44}{12} \qquad (9)$$

$$Seq = \left( \sum_{l=1}^{L} Area_l \times \sum_{i=36}^{40} f_{TECS}(i) + \sum_{m=1}^{M} Area_m \times \sum_{i=31}^{40} f_{TECS}(i) + \sum_{n=1}^{N} Area_n \times \sum_{i=21}^{40} f_{TECS}(i) \right) \times \frac{44}{12} \qquad (10)$$

Where $Area_{2016-1996}$ represents the total feasible reforestation extent from 1996 to 2016; $f_{AGC}$, $f_{BGC}$, $f_{SCS}$ represent the best-fit carbon accumulation model of aboveground biomass carbon, belowground biomass carbon, and sediment carbon pools for mangrove reforestation simulated above, respectively. $L$, $M$, and $N$ were the number of countries with 5-year, 10-year, and 20-year restoration periods under assumptions listed in scenario 4. We also calculated the $CO_2$-eq sequestration potential for afforestation by assuming the same potential afforestation area as for reforestation with carbon accumulation models. All geoinformation processing was executed using QGIS (version 3.18.2, https://www.qgis.org/) and packages in R (dplyr (version 1.0.9), tidyr (version 1.2.0), rstatix (version 0.7.0), ggpmisc (version 0.4.6), ggpubr (version 0.4.0), ggrepel (version 0.9.1), ggalluvial (version 0.12.3), reshape2 (version 1.4.4), introdataviz (version 0.0.0.9003), sf (version 1.0.7), rnaturalearth (version 0.1.0), rnaturalearthdata (version 0.1.0), nlraa (version 1.2), and agricolae (version 1.3.5)).

### Reporting summary
Further information on research design is available in the Nature Portfolio Reporting Summary linked to this article.

### Data availability
All data collected through literature search are available at https://doi.org/10.6084/m9.figshare.21032425; data supporting Supplementary Fig. 8 are available at https://doi.org/10.6084/m9.figshare.21032764; mangrove area dataset (Global mangrove watch 1996 and 2016) is available at https://data.unep-wcmc.org/datasets/45; assumption and calculation process of climate mitigation potential from global

mangrove reforestation under biophysical constraint is available via https://doi.org/10.6084/m9.figshare.21032644; mangrove losing area in each country and its exclusive economic zone are available via https://doi.org/10.6084/m9.figshare.21769763; WorldClim2.0 dataset are available via https://www.worldclim.org/data/worldclim21.html; The union of world country boundaries and its exclusive economic zone are available via https://www.marineregions.org/downloads.php. Source data of figures and tables are provided in the Source Data file with this paper.

### Code availability
Codes for creating each figure are available from https://doi.org/10.5281/zenodo.7556422.

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

## Acknowledgements
This analysis was supported in part by the National Key R&D Program of China (2019YFA0606604, G.L.), Shenzhen City's University Stable Support Program (WDZC2020819173345002, G.L.), Tsinghua University Initiative Scientific Research Program (20223080041, W.L.), Preparation and Pilot Research Program of Hainan International Blue Carbon Research Center (46000022T000000154465, S.L.), and the US Geological Survey Climate R&D Program (K.W.K.). Any mention of trade, product, or firm names is for descriptive purposes only and does not imply endorsement by the US Government.

## Author contributions
S.S., G.L., W.L., and Y.D. conceived the study; S.S. collected data, conducted the analyses, and wrote the first manuscript; Y.M. provided research data collected from the previous study. All co-authors discussed the results, gave suggestions for further analyses, and contributed significantly to the revision of the manuscript.

## Competing interests
The authors declare no competing interests.
