## [Peer Review File · Nature Communications]

Reviewers' Comments:

Reviewer #1:

Remarks to the Author:

The manuscript deals the estimated carbon budget in mangrove reforestation and afforestation by using published data globally. The data analysis mainly focuses on comparison of carbon budget between the two restoration scenarios. It gives merit for mangrove rehabilitation in tropic and subtropic regions based on purpose of enhancing carbon sequestration potential.

The topic of manuscript presents "mitigation of climate change", however the analysis has not been done for the effect of common climatic factors, i.e., annual mean precipitation (MAP), annual mean air temperature (MAT) on the magnitude of carbon budget in mangrove restoration or afforestation. While these mentioned factors are said to affect the mangrove growth and consequently its ecological productivity. Therefore, I would like to suggest the refine the available data which reports the factors for re-analysis. This re-analysis will separately explain the carbon budget regarding to the restoration pathway and casual climatic factors.

Also, I would like the authors to exclude the data of Hawaii, USA (Line 91-94) from the analysis because Hawaii is not an original mangrove habitat. Its carbon budget cannot be compared with the others original habitat in the tropic and subtropic regions.

Introduction:

1. The mangrove restorations areas were not consistent (Line 46-47). The authors gave examples of the locations as continental part (Southeast Asia), countries (Columbia, Panama, China), and specific place (the Arabian Gulf). Please unify them by using mangrove regions.
2. "other forest types" in Line 63 means terrestrial forests? Please make it clear.
3. "amount of carbon" in Line 67 should be clearly stated as necromass or organic matter which is prominent carbon pool in the mangrove ecosystem.
4. I think that mention of advantage and disadvantage of both restoration scenarios should be briefly appeared in the section.

Results and discussions:

1. Line 137 "prevent erosive losses"; this is an outstanding role of mangrove roots in a spatial of root density which reflects the belowground carbon of mangrove. However, the authors attributes to rapid development of aboveground carbon biomass and sediment carbon (Line132-135). Please rewrite how relate between the "rapid development of aboveground carbon biomass and sediment carbon" and "prevent erosive losses" resulted by climate change.
2. Line 162-163, I supposed that TOC represents Total Organic Carbon. Please confirm.
3. Line 202-206, the authors discuss leaf litter quality (C:N) affecting the leaf litter decomposition. However, a characteristic of succulent leaf of *Rhizophora* spp. is also like to enhance the microbial decomposition. I recommend the authors to mention this in the discussion.
4. As I mentioned earlier that data analysis should be done for the climatic factors and carbon budget separately by the reforestation/afforestation. Consequently, Figure 3 is necessary to be revised.
5. Line 301-303, the aboveground carbon was stated in a comparison between the two restoration scenarios in the tropical regions. But the root biomass is also important due to the low top/root biomass in the mangrove forest. Please revise this part to highlight the belowground biomass and carbon of the mangrove ecosystem.
6. Line 315-320, the authors convince that different freshwater inputs and salinity relief have an influence on seedling growth in restored sites such as abandoned maricultural ponds. Nevertheless, I suppose that edaphic factors (i.e., soil bulk density, soil texture and aeration) play significant roles on the seedling establishment and root growth. Root growth and expansion may be restricted in clay pan soil resulted by sedimentation in abandoned maricultural ponds.

Reviewer #2:

Remarks to the Author:

The manuscript "Mangrove reforestation provides greater blue carbon benefit than afforestation for

mitigating global climate change" is interesting and a nicely assembled manuscript. The topic of the paper is of interest and worthy of publication. However, there are issue that should be clarified before further considerations may be given.

The major concern I have is with collecting 1 meter soil profiles. It would take several hundred years for mangroves to accumulation this amount of carbon so not sure how this relates the to up to 40 year restoration and afforestation comparisons outlined here. A better explanation should be given to justify these soil depths. For example, the deforested sites will likely already contain several (perhaps hundreds) of years of soil carbon accumulation from when the forest was intact, which is not the case for the afforested sites. Perhaps it would be more of interesting to look at carbon accumulation rate differences from the onset of reforestation and aforestaon, which would make more sense and a better direct comparison. Either way more attention needs to given and better explanations should be provide when addressing the soil carbon comparison between reforestation and afforestation sites as this was a major point of difference in carbon stocks when comparing these two methods.

Another issue I found in the manuscript is dealing with the cause of the degradation or deforestation:

Line 49: Referring to "fairly recent degradation or deforestation", are the causes of this degradation or deforestation not important here? If not, are we to assume these forests will be degraded or deforested in the near future?

Line 67: "historical death" again, seems important to address what caused these deaths otherwise would not make a lot of sense to reforest.

Minor issues

Line 114, this subtitle seems presumptive, as the reader has not even seen results yet.

Figure 3, please add titles to the X axes.

Reviewer #3:

Remarks to the Author:

Mangrove reforestation provides greater blue carbon benefit than 2 afforestation for mitigating global climate change

Referee Report

The manuscript addresses the relevant questions of whether mangrove reforestation or afforestation activities provide larger carbon sequestration benefit. The topic holds great promise in guiding and prioritizing future mangrove conservation activities and investments. The breadth of analysis (370 restoration sites globally) is impressive and of broad enough scope to be useful in contributing to the conversation on global mangrove conservation. The paper is well motivated and distinguishes itself in focuses on mangrove conservation prioritization as it relates to the past land use of target areas. The contribution of the paper is muted, however, by its idealistic simulations of potential carbon storage from reforestation vs. afforestation activities. Of course, it is unsurprising that reforestation activities have greater potential as described well by the authors. More relevant for global policy and mangrove conservation action is whether such projects are feasible in areas viable for reforestation vs. afforestation. At present, the authors make no effort to understand the local opportunity cost of mangrove reforestation in areas primed for reforestation. Nor is attention paid to the opportunity cost of mangrove afforestation in areas primed for afforestation. This being said, the paper is still worthy of future publication in *Nature Communications* with a major revision that recognizes the limitations of the paper's contribution in global mangrove conservation policy. An ideal revision would refine the current analysis to areas with an exhibited desire to engage in reforestation activities (rather than excluding only degraded and developed areas).

Line 37: replace "same" with "the same"

Line 45-47: revise sentence structure "covering, e.g.,".

Lines 139-142: The claim "These data even...restoration sites" requires more justification and explanation for an audience as general at Nature Communications. Given the heavy framing around carbon storage, please provide additional context and framing around the related topic of wood production before making this leap, which is currently assumed to be understood by readers.

I must defer to my colleagues and other reviewers in the natural and physical sciences to assess the findings presented in Fig:2-3.

Lines 255-257: This headline number is impressive and worth presenting. However, it is idealistic to think that such comprehensive mangrove reforestation activities are feasible. In the same way that the authors exclude eroded and developed locations, it would make sense to conduct a more nuanced and pragmatic exclusion of areas that have such a higher opportunity cost of reforestation that they are irrelevant. Of course, many mangrove areas that were recently deforested experienced extraordinary pressure from land use change and reversing that pressure, in many cases, is unlikely. In other words, even though afforestation activities provide less carbon storage protentional... they might be easier and more viable targets relative

to areas that are socially prohibitive to reforest. Can you also present a refined number that focuses on areas with proposed or existing mangrove reforestation efforts and/or high likely and/or social and economic suitability for reforestation?

Lines 262-266: The same argument could be made for reforesting mangroves that will replace built and public services that arose from those deforestation activities with ecosystem services such as carbon storage. This is why the subanalysis suggested in my previous comment is important as not to focus on the idealistic case of full reforestation without recognizing that many of these lands have high local post-deforestation value that would render their reforestation cost prohibitive. Note the distinction in lines 266-268 “Furthermore, ... new mangrove habitat” the authors are referring to the cost of actual reforestation activities rather than the opportunity costs of land conversion that will be experienced by local communities that drove initially the deforestation of these mangroves. Therefore, it is important to present evidence specifically for areas that are primed to engage in such reforestation activities.

Reviewer #1 (Remarks to the Author):

The manuscript deals the estimated carbon budget in mangrove reforestation and afforestation by using published data globally. The data analysis mainly focuses on comparison of carbon budget between the two restoration scenarios. It gives merit for mangrove rehabilitation in tropic and subtropic regions based on purpose of enhancing carbon sequestration potential.

The topic of manuscript presents “mitigation of climate change”, however the analysis has not been done for the effect of common climatic factors, i.e., annual mean precipitation (MAP), annual mean air temperature (MAT) on the magnitude of carbon budget in mangrove restoration or afforestation. While these mentioned factors are said to affect the mangrove growth and consequently its ecological productivity. Therefore, I would like to suggest the refine the available data which reports the factors for re-analysis. This re-analysis will separately explain the carbon budget regarding to the restoration pathway and casual climatic factors.

Re: Many thanks to the reviewer’s positive comments on our work. We agree with the reviewer that climate factors may influence the mangrove growth as well as carbon storage during mangrove restoration, especially in a global-scale analysis. We thus extracted these climate factors (i.e., mean annual precipitation, MAP, and mean annual temperature, MAT, as the reviewer suggested) from the WorldClim2.0 dataset using the longitude and latitude of each restoration site (see details in Methods, Line 498–501). A linear mixed model was then used to test the influence of restoration pathway and casual climate factors on the aboveground and belowground biomass carbon accumulation. We included age, restoration pathway, MAT, MAP and their interactions as fixed factors, with restoration region as a random factor (see details in Supplementary Text 1).

The results show that both restoration pathways and MAP can influence the aboveground biomass carbon accumulation over time ($\log(\text{Age}) \times \text{restoration pathways}$, $P \leq 0.05$; $\log(\text{Age}) \times \text{MAP}$, $P \leq 0.05$, Supplementary Table S2), while belowground biomass carbon is only influenced by MAP. We further classified mangrove restoration

sites into three groups with different precipitation levels of 500–1500, 1500–2500 and 2500–3500 mm yr⁻¹ of MAP. For both restoration pathways, greater aboveground biomass carbon accumulation always occurs in regions with higher amount of precipitation (except for afforestation sites with MAP of 2500–3500 mm yr⁻¹). Meanwhile, within each MAP interval, mangrove reforestation achieved a greater aboveground biomass carbon accumulation than afforestation (Supplementary Fig. S4). In short, MAP and restoration pathways jointly controlled the carbon sequestration of aboveground biomass during mangrove restoration at the global scale.

In our database, there is no significant difference of MAP between mangrove reforestation and afforestation sites (Wilcoxon test, $W=23264$, $P>0.05$), although MAP in the reforestation sites is more evenly distributed, and MAP in the afforestation sites mainly concentrates ~2000 mm yr⁻¹ (Supplementary Fig. S4). The interaction between MAP and restoration pathways ($\log(\text{Age}) \times \text{restoration pathways} \times \text{MAP}$, $P>0.05$; Supplementary Table S2) is non-significant, suggesting that carbon accumulation during mangrove reforestation and afforestation was influenced by MAP to a similar extent. Therefore, we didn't provide the regrowth curve for each MAP intervals due to the limited data for regression.

We have added these statements about the effect of climate factors and restoration pathways in the revised manuscript (Line 66–69, 170–176, 354–359 in the main text, and supplementary Text 1).

Table S2. Impacts of climate factors and restoration types on carbon densities over time examined by linear mixed models. AGC: aboveground biomass carbon; BGC: belowground biomass carbon. Restoration pathways: reforestation and afforestation; MAT: mean annual temperature; MAP: mean annual precipitation. AIC: Akaike information criterion. AIC-log: model with log-transformed age; AIC-original: model with age. A lower AIC value indicates a better model fit. Significant variables ($P \leq 0.05$) are shown in bold.

Carbon pool	Variable	P -value	AIC-log	AIC-original
	Log (Age)	<0.001		
	Restoration pathways	0.64		
	MAT	0.41		
	MAP	0.02		
	Log (Age) × Restoration pathways	0.01		
AGC	Log (Age) × MAT	0.68	2953.90	2968.73
	Log (Age) × MAP	<0.001		
	Restoration pathways × MAP	0.98		
	Restoration pathways × MAT	0.11		
	Log (Age) × Restoration pathways × MAP	0.79		
	Log (Age) × Restoration pathways × MAT	0.08		
	Log (Age)	<0.001		
	Restoration pathways	0.72		
	MAT	0.35		
	MAP	0.003		
	Log (Age) × Restoration pathways	0.39		
BGC	Log (Age) × MAT	0.70	1903.59	1921.04
	Log (Age) × MAP	0.09		
	Restoration pathways × MAP	0.05		
	Restoration pathways × MAT	0.89		
	Log (Age) × Restoration pathways × MAP	0.11		
	Log (Age) × Restoration pathways × MAT	0.20		

Fig. S4 Comparison of carbon densities between mangrove reforestation and afforestation sites in different precipitation intervals over a 40-year period. a, distribution of mean annual precipitation (MAP) in mangrove reforestation and afforestation sites. Vertical lines from left to right represents MAP levels of 500, 1500, 2500 and 3500 mm yr^{-1} , which are used as the boundary values for different precipitation groups. Aboveground (**b**) and belowground (**c**) carbon density changes

with mangrove ages (x-axis) for each precipitation group (different colors). The x-axis represents different age groups (0–5, 5–10, 10–15, 15–20, and 20–40 years). For the box plot, the center line and the top and bottom of the box represent the median and the interquartile range (25th percentile and 75th percentile). The whiskers represent the minimum and maximum limits, and the outliers are represented by dots. Mean carbon density in each age group and each MAP group is represented by a circle and connected by a line. Sample size (n) of each group is showed at the bottom. Significance level of difference between reforestation and afforestation within each age group is calculated by the Kruskal–Wallis test.

Also, I would like the authors to exclude the data of Hawaii, USA (Line 91-94) from the analysis because Hawaii is not an original mangrove habitat. Its carbon budget cannot be compared with the others original habitat in the tropic and subtropic regions. Re: Thank you for bringing this up. The mangrove habitat reported in the original paper was introduced to Hawaii about 70 years ago¹. We incorporated it into our database because it could be regarded as ‘afforestation’ according to our classification (establishing mangroves where they have not previously colonized). However, we did not include the data of Hawaii in our further analysis in Fig. 2–4 because these figures only considered mangrove plots within 40 years.

Introduction:

1. The mangrove restorations areas were not consistent (Line 46-47). The authors gave examples of the locations as continental part (Southeast Asia), countries (Columbia, Panama, China), and specific place (the Arabian Gulf). Please unify them by using mangrove regions.

Re: Thank you for your suggestion. We have changed the sentence into “Since the 1970s, mangrove restoration projects have been initiated in regions such as Southeast Asia, East Asia and South America^{2,3}” (Line 43–44).

2. “other forest types” in Line 63 means terrestrial forests? Please make it clear.

Re: As suggested, we changed it to ‘terrestrial forests’ (Line 70). We referred to two studies on temperate forests⁴ and tropical dry forests⁵ here.

3. “amount of carbon” in Line 67 should be clearly stated as necromass or organic matter which is prominent carbon pool in the mangrove ecosystem.

Re: We have changed it into “organic matter” (Line 74).

4. I think that mention of advantage and disadvantage of both restoration scenarios should be briefly appeared in the section.

Re: As suggested, we have added advantage and disadvantage of reforestation and afforestation, as follows (Line 50–56):

Generally, the trade-offs in choosing between reforestation and afforestation relate to the recovery efficiency and the sociopolitical complexities of land tenure^{6,7}. Reforestation may avoid conflicts of converting other vulnerable ecosystems to mangroves but still face tenurial problems, especially for the abandoned maricultural ponds. Afforestation sites (e.g., mudflats) may have lower land cost, but the survival rate of seedlings is often low because of the inappropriate hydrodynamic conditions. Quantification of possible differences in ecosystem benefits such as carbon sequestration between mangrove reforestation and afforestation is thus crucial for formulating policies and plans that specify mangrove restoration protocols to enhance blue carbon potential and help with mitigating global climate change.

Results and discussions:

1. Line 137 “prevent erosive losses”; this is an outstanding role of mangrove roots in a spatial of root density which reflects the belowground carbon of mangrove. However, the authors attributes to rapid development of aboveground carbon biomass and sediment carbon (Line132-135). Please rewrite how relate between the “rapid development of aboveground carbon biomass and sediment carbon” and “prevent erosive losses” resulted by climate change.

Re: Sorry for the confusion. As suggested, we rewrote this sentence to make it clear

(Line 146–148): Mangrove reforestation is likely a more effective strategy in support of climate change mitigation actions, which may simultaneously prevent further losses of in-situ sediment carbon existing before the reforestation action.

Mangrove roots can also help avoid erosive loss and microbial mineralization of old sediment carbon through sediment trapping and new carbon input into sediment⁸. Although the mangrove reforestation and afforestation have a similar carbon accumulation rate in belowground biomass, the amount of sediment carbon existing before reforestation was nearly twice of that before afforestation (Fig. 2c). The avoided carbon loss from sediment, therefore, would be greater for mangrove reforestation than afforestation.

2. Line 162-163, I supposed that TOC represents Total Organic Carbon. Please confirm.

Re: We have changed it to “total organic carbon” (Line 184).

3. Line 202-206, the authors discuss leaf litter quality (C:N) affecting the leaf litter decomposition. However, a characteristic of succulent leaf of *Rhizophora* spp. is also like to enhance the microbial decomposition. I recommend the authors to mention this in the discussion.

Re: Thank you for your suggestion. We added discussion of this point in Line 226–235: Compared with non-succulent leaves, succulent leaves usually store more nitrogen content⁹, which is likely to enhance higher microbial decomposition. However, mangrove leaves also have a high concentration of tannins to deter crabs from leaf-herbivory¹⁰. Despite the succulent characteristic of *Rhizophora* spp. leaves (species dominating reforestation projects globally), its higher tannin and lignocellulose content may hamper litter decomposition, especially when compared with the *Kandelia* spp. and *Sonneratia* spp. (species widely used in the afforestation projects, at least in the Indo-Pacific region) with higher content of fatty acids and amino acids¹¹⁻¹³, as observed by some litter bag incubation studies^{11,14}.

4. As I mentioned earlier that data analysis should be done for the climatic factors and

carbon budget separately by the reforestation/afforestation. Consequently, Figure 3 is necessary to be revised.

Re: Please see the reply above for the new analysis of the climatic factors. Following the suggestion, we also added the influence of climate factors on aboveground biomass for both reforestation and afforestation into Figure 3. The results show that aboveground biomass accumulation at both reforestation and afforestation sites can be promoted by an increase of mean annual precipitation (MAP) (Fig. 3a, b). However, the similar distributions of MAP in the two restoration pathways and their insignificant interactions indicated that MAP is not the main reason for the differences in carbon sequestration between reforestation and afforestation. We added these statements into Line 170–176.

Fig. 3 Relationship of aboveground biomass carbon density with mean annual precipitation and sediment properties for mangrove restoration. a, c, e, g, Relationship of aboveground biomass carbon (AGC) density with mean annual precipitation (MAP), sediment total organic carbon content, total nitrogen content,

and porewater salinity. Lines in **a**, **c**, **e** and **g** represent the ordinary least squares regression. psu refers to practical salinity units. **b**, **d**, **f**, **h**, Comparisons of MAP and sediment properties between reforestation and afforestation sites. Note that some sites with belowground biomass or sediment organic carbon density data but without aboveground biomass data are also included in **b**, **d**, **f** and **h**. For each box plot, individual data points are shown as circles, the center line and the top and bottom of the box represent the median and the interquartile range (25th percentile and 75th percentile). The whiskers represent the minimum and maximum limits. Sample size (n) of each group is shown at the bottom of each boxplot. Significance level of difference between reforestation and afforestation is calculated by the Wilcoxon test.

5. Line 301-303, the aboveground carbon was stated in a comparison between the two restoration scenarios in the tropical regions. But the root biomass is also important due to the low top/root biomass in the mangrove forest. Please revise this part to highlight the belowground biomass and carbon of the mangrove ecosystem.

Re: Thank you for your suggestion. In the original manuscript, we used climate zones (tropical and subtropical regions) to explain the interactive effects of climate factors and restoration pathways. The results showed that aboveground biomass in tropical regions after 15-year growth was greater in reforestation than afforestation. Following your first comment, we restructured this part of discussion by introducing mean annual precipitation (MAP) and temperature (MAT) into our analysis (see detailed methods in Supplementary Text 1). The results show that belowground biomass is influenced by MAP but not by restoration pathways (Supplementary Table S2, Fig. S4). Therefore, the original sentence was removed in the revised version, and the importance of belowground biomass was highlighted in the new analysis with climate factors (Line 359–362).

6. Line 315-320, the authors convince that different freshwater inputs and salinity relief have an influence on seedling growth in restored sites such as abandoned maricultural

ponds. Nevertheless, I suppose that edaphic factors (i.e., soil bulk density, soil texture and aeration) play significant roles on the seedling establishment and root growth. Root growth and expansion may be restricted in clay pan soil resulted by sedimentation in abandoned maricultural ponds.

Re: Thanks for these great points. We reorganized the sentence to make it clearer (Line 367–371): In an estuarine tidal flat actively undergoing sedimentation, *Kandelia* spp. afforested for 3–6 years exhibited similar growth characteristics as a nearby reforested site in abandoned maricultural ponds (Supplementary Fig. S9), probably because of relative greater riverine freshwater inputs and salinity relief to the afforested tidal flat.

We also added discussion about sediment edaphic properties in Line 371–379: Sediment textures of abandoned ponds usually depend on the local hydrogeomorphic conditions associated with their original mangrove setting¹⁵. For example, the substrate of abandoned fish ponds is mainly gravelly sand in sandstone coastal regions¹⁶, but mainly clay-loam or silt clay-loam in riverine-influenced areas¹⁷. These variations of edaphic and hydrological properties could lead to plastic but species-specific dynamics of the above- and belowground biomass^{18,19}. Therefore, more detailed information on classifications within each restoration pathway and on sediment edaphic properties at the site scale can be expected to inform more accurate selection of restoration sites.

Reviewer #2 (Remarks to the Author):

The manuscript “Mangrove reforestation provides greater blue carbon benefit than afforestation for mitigating global climate change” is interesting and a nicely assembled manuscript. The topic of the paper is of interest and worthy of publication. However, there are issue that should be clarified before further considerations may be given.

Re: We thank the reviewer for the positive comments on our manuscript, and we have revised the manuscript following the constructive and helpful suggestions from the reviewer.

The major concern I have is with collecting 1 meter soil profiles. It would take several hundred years for mangroves to accumulation this amount of carbon so not sure how this relates to up to 40-year restoration and afforestation comparisons outlined here. A better explanation should be given to justify these soil depths. For example, the deforested sites will likely already contain several (perhaps hundreds) of years of soil carbon accumulation from when the forest was intact, which is not the case for the afforested sites. Perhaps it would be more of interesting to look at carbon accumulation rate differences from the onset of reforestation and afforestation, which would make more sense and a better direct comparison. Either way more attention needs to be given and better explanations should be provided when addressing the soil carbon comparison between reforestation and afforestation sites as this was a major point of difference in carbon stocks when comparing these two methods.

Re: We thank the reviewer for this great suggestion. The period of 40 years is mainly for the biomass growth of mangrove restoration, we agree that 40 year is relatively short for the long-term carbon accumulation in the 1 m soil depth. There are two main reasons for using the 1 m soil depth in our study. (1) During mangrove stand development, sediment organic matter accumulation originates from both autochthonous and allochthonous sources, in which refractory root litter input plays a vital role^{8,20}. Despite mangrove root biomass decreasing with sediment depth, 80% of root biomass was found in the top meter of sediment²¹ with nearly 30% of living roots in the deeper root zone (>45 cm)²². Therefore, using 1 meter soil profile could reflect the impact of root

mass input on the sediment carbon dynamic during mangrove restoration; (2) Although some protocols for mangrove carbon stock measurements have been promoted in recent years^{23,24}, stratification criteria for 1 m depth sediment varied widely among studies (i.e., by 10 cm²⁵, 15 cm²⁶, 20 cm²⁷, and the entire 1 m depth²⁸). For the data consistency with established blue carbon assessment protocol, 1 m depth soil profile was thus selected in our analysis (Line 486–487).

As the reviewer pointed out, the initial sediment carbon stock before mangrove restoration could be higher at the reforestation sites than the afforestation sites, because of the long-term carbon accumulation before deforestation at the reforestation sites. Following the reviewer's suggestion, we added a new pair-site analysis to remove the differences in the initial soil carbon before restoration between reforestation and afforestation sites, in addition to the original analysis of restoration pathways (Fig. 2c). Specifically, we collected the sediment properties in the control area without mangrove rehabilitation as a paired site of restoration using the space-for-time method (e.g., abandoned ponds before mangrove reforestation; mudflats before mangrove afforestation) (Line 449–452). The difference in sediment carbon densities between the restoration site and the paired control site (without restoration) was considered as the carbon change induced by mangrove restoration. We acknowledge that some uncertainties exist in the space-for-time method because it is difficult to find a 'perfect' control site, which may partly explain the varying, even negative, carbon stock increments in some age groups (Supplementary Fig. S2). To minimize the uncertainties, we corrected the negative values if the carbon accumulation rate was measured simultaneously by the rot surface elevation tables-marker horizon²⁹ and recalculated the carbon density increments via multiplying the carbon accumulation rate by its restoration duration (Line 515–530).

After excluding the initial sediment carbon storage before restoration, the sediment carbon increments since the time of reforestation are still higher than those for afforestation, although the increments for reforestation tend to be smaller as mangrove growing older (Supplementary Fig. S2) (Line 135–139). Since the results of the pair-site analysis are consistent with the previous analysis using the total sediment carbon

density, we maintained the use of sediment carbon density in Fig. 2 and Fig. 4 in the main text. It is because the total sediment carbon density contains additional information (i.e., the initial carbon stock before restoration) than the increments. The larger initial sediment carbon stock at the reforestation sites than the afforestation sites further emphasize the importance of reforestation on avoiding further sediment carbon loss.

Fig. S2 Sediment carbon increments from the onset of mangrove reforestation and afforestation. Sediment carbon density increments are calculated as the difference between each restoration site and its paired control site. The x-axis represents different age groups (0–5, 5–10, 10–15, 15–40 years). For the box plot, the center line and the top and bottom of the box represent the median and the interquartile range (25th and 75th percentile). The whiskers represent the minimum and maximum limits, and the outliers are represented by dots. Mean carbon density in each age group and each MAP group is represented by a circle and connected by a line. Sample size (n) of each group is showed at the bottom. Significance level of difference between reforestation and afforestation within each age group is calculated by the Wilcoxon test.

Another issue I found in the manuscript is dealing with the cause of the degradation or deforestation:

Line 49: Referring to “fairly recent degradation or deforestation”, are the causes of this degradation or deforestation not important here? If not, are we to assume these forests will be degraded or deforested in the near future?

Re: Thank you for bringing it to our attention. We changed our description in Line 49 to make it clear. The driving factors of degradation and deforestation can be both natural- and anthropogenic-driven, while in our database, it is mostly anthropogenic-driven. To avoid repetition, these driving factors were described in details in Methods (Line 421–425). In our database, reforestation occurs in (1) abandoned agricultural/aquacultural sites built previously by excavating mangrove forests, (2) clear-felled mangrove lands after wars, timber harvest and silvicultural management, and (3) mangrove forests with mortality due to spraying of defoliant and hydrological alteration caused by the construction of embankments. Reforestation refers to silvicultural action establishing seedlings where they previously occurred (within 50 years) by definition, that is, sites that have experienced fairly recent degradation or deforestation^{30,31}. We compared the carbon densities of reforested mangroves among sites with different causes of degradation/deforestation, and no significant difference is found (Supplementary Fig. S10, Line 425–428).

For those reforested mangroves, we assumed they would be protected and conserved by local governments and non-government organizations, so that there will not be human-driven degradation or deforestation in the near future. However, we acknowledge that a fraction of mangrove reforestation is managed for wood production, which means logging would happen at a certain interval after reforestation at these sites. For these logging sites, we used their reported measurements after clear-cut, such as 0-, 5-, 10-, 15- and 25-year post-harvest sites in Sundarbans, Bangladesh²⁹. On the other hand, the future occurrence of natural-driven deforestation (e.g., cyclones) is difficult to predict, and thus not considered in our study (Line 428–436).

Fig. S10 Comparison of mangrove carbon densities after 15-40 years of restoration among sites with different causes of degradation/deforestation. For each box plot, individual data points are shown as circles. The center line and the top and bottom of the box represent the median and the interquartile range (25th percentile and 75th percentile). The whiskers represent the minimum and maximum limits. Sample size (n) of each group is showed at the bottom. Significance level of difference among age groups is calculated by the Kruskal–Wallis test combined with the Bonferroni adjusted post hoc Dunn test. Groups sharing the same letter were not significantly different at level of 0.05.

Line 67: “historical death” again, seems important to address what caused these deaths otherwise would not make a lot of sense to reforest.

Re: Please see the reply above. We also add the detailed reasons for the historical death in Line 72–75: In regions that suffered historical hypersalination-driven death after road levee construction, restored mangroves could store greater amounts of organic matter and nutrients in sediments than currently conserved mangrove area²⁵.

Minor issues

Line 114, this subtitle seems presumptive, as the reader has not even seen results yet.

Re: Thank you for your suggestion. We have changed it to ‘Mangrove carbon accumulation after reforestation and afforestation’ (Line 120). The second subtitle was also changed to ‘Factors controlling mangrove carbon accumulation’ for the same

reason (Line 169).

Figure 3, please add titles to the X axes.

Re: Revised accordingly.

Reviewer #3 (Remarks to the Author):

The manuscript addresses the relevant questions of whether mangrove reforestation or afforestation activities provide larger carbon sequestration benefit. The topic holds great promise in guiding and prioritizing future mangrove conservation activities and investments. The breadth of analysis (370 restoration sites globally) is impressive and of broad enough scope to be useful in contributing to the conversation on global mangrove conservation. The paper is well motivated and distinguishes itself in focuses on mangrove conservation prioritization as it relates to the past land use of target areas. The contribution of the paper is muted, however, by its idealistic simulations of potential carbon storage from reforestation vs. afforestation activities. Of course, it is unsurprising that reforestation activities have greater potential as described well by the authors. More relevant for global policy and mangrove conservation action is whether such projects are feasible in areas viable for reforestation vs. afforestation. At present, the authors make no effort to understand the local opportunity cost of mangrove reforestation in areas primed for reforestation. Nor is attention paid to the opportunity cost of mangrove afforestation in areas primed for afforestation. This being said, the paper is still worthy of future publication in Nature Communications with a major revision that recognizes the limitations of the paper's contribution in global mangrove conservation policy. An ideal revision would refine the current analysis to areas with an exhibited desire to engage in reforestation activities (rather than excluding only degraded and developed areas).

Re: We appreciate the positive and constructive comments from the reviewer. We agree that local opportunity cost of mangrove restoration is critical for the feasibility of the restoration project and thus more related to the global mangrove conservation policy.

In the original manuscript version, we only considered the biophysical constraints and assumed that feasible reforestation locations were in regions with (1) mangrove dieback after extreme weather events, (2) mangrove clearing-cutting, and (3) mangroves converted into commercial protein production plots, such as aquaculture and agriculture³². Especially in areas converted into lands with commercial profit, mangrove restoration leads to a high land opportunity cost. Therefore, in addition to the biophysical constraint, we also considered the economic constraint and excluded regions with high reforestation costs, following the methods for many other nature-based solutions^{33,34}.

We considered two main components (implementation cost and opportunity cost) of restoration cost at the national scale due to lack of data at finer scales. The first component, restoration implementation cost, includes capital, maintaining, and in-kind cost. We extracted it from a recent review of global mangrove restoration cases³⁵, which shows that the total restoration cost is significantly lower in developing countries than developed countries. In addition, we further classified these restoration cases into reforestation and afforestation based on whether mangrove previously grew. Cases reporting a synthesis of restoration pathways in the review³⁵ were not used in our study. Therefore, we derived four values of average direct restoration implementation cost specifically for each combination of one pathway (reforestation or afforestation) and one country state (developed or developing) (Supplementary Table S4). We then applied these four cost values to country-specific values based on their gross domestic product per capita (GDP per capita) in 2021³⁶ (equation (6)).

$$Cost_i = \frac{GDP_i}{\sum_{i=1}^n GDP_i/n} \times Average Cost \quad (6)$$

GDP_i represents the GDP per capita in the i_{th} country; $n = 94$ for developing countries and $n = 10$ for developed countries.

The second component, country-level opportunity cost, was derived directly from a global dataset of opportunity costs for mangrove restoration by Jakovac, et al.³⁷. In their study³⁷, they estimated the opportunity costs with the net present value of future profits that would be foregone when restoring agriculture and pastures back to

mangroves. Opportunity costs in some mangrove-holding countries that are not considered by Jakovac, et al.³⁷ were also calculated by equation (6) with the average opportunity cost weighted by their GDP per capita (Supplementary Table S4).

We defined the sum of implementation cost and opportunity cost as the total economic cost for reforestation in regions with commercial profits, while only the implementation cost was used for reforestation in regions with mangrove clearing-cut or mortality. All the costs were converted into 2021 U.S. dollars.

The CO₂-eq sequestration potential (Mg CO₂-eq ha⁻¹) under mangrove reforestation was calculated as the sum of carbon density increments of AGC, BGC, and SCS from the initial baseline (when age was 0), which are predicted by the corresponding growth models and confidence intervals. We used a period of 40 years (i.e., up to 2060) to assess carbon sequestration potential. We then calculated the cost per unit CO₂-eq (i.e., mitigation cost, Supplementary Fig. S7). To evaluate the economic constraints on reforestation, we assumed that the mitigation cost should be lower than the three commonly used cost thresholds (i.e., \$10, \$50 and \$100 Mg CO₂-eq⁻¹), otherwise it would be economically infeasible. The cost threshold of \$10 Mg CO₂-eq⁻¹ represents the average current carbon prices in crediting mechanisms covering forestry, while \$50 Mg CO₂-eq⁻¹ and \$100 Mg CO₂-eq⁻¹ represent a median and high-end price for limiting global warming to below 2 °C above pre-industrial levels, respectively³⁸. These price thresholds were also benchmarked to other nature-based solutions in previous studies^{33,34,39}. We acknowledge that the debate over land tenure for individual projects may hinder mangrove restoration⁶, even if they are not biophysically and economically constrained. However, this was not taken into account in our analysis because the global data of land tenure are not available and it is beyond the scope of this study.

At a cost threshold of \$100 Mg CO₂-eq⁻¹, the land area available for mangrove reforestation is 564,073 ha, compared to 614,472 ha with only biophysical constraints but without economic constraints. Assuming all these areas to be restored within 1 year (Scenario 1 in Supplementary Table S3), the cumulative carbon sequestration over the next 40 years is 630.8 (572.7–688.9) Tg CO₂-eq, which is about 234 Tg CO₂-eq higher

than afforestation (Fig. 4e, '1 year-completed restoration' scenario, see Methods). Indonesia, Mexico, and Myanmar will provide the greatest carbon mitigation potential through reforestation (188.2, 110.1, and 61.2 Tg CO₂-eq over 40 years, respectively) (Fig. 4e). Under other scenarios with slower rates of restoration area, i.e., reforesting all these areas within a 5-year and 10-year period (Scenario 2 and 3 in Supplementary Table S3), the cumulative carbon sequestration potential during 2021–2060 is 629.1 (575.5–682.7) and 625.7 (579.7–671.8) Tg CO₂-eq, respectively, and it reaches 612.3 (584.4–640.2) Tg CO₂-eq in the scenario with varying rates of restoration area across countries (Scenario 4 in Supplementary Table S3). At the cost thresholds of \$50 and \$10 Mg CO₂-eq⁻¹, available restoration area in 2021 reduces to 563,523 ha and 224,854 ha, and the cumulative carbon sequestration potential during 2021–2060 ranges from 244.7 (232.7–256.7) to 630.2 (572.1–688.3) Tg CO₂-eq under various restoration rates (Supplementary Table S3).

Despite of the reduced climate mitigation potential caused by physical and economical constraints, mangrove restoration can also bring co-benefits with higher ecosystem values such as coastal protection, biodiversity improvements, and fisheries services³⁸. These co-benefits were not included in our estimates, but they can also help reduce the economic cost of restoration. Moreover, only regions with mangrove deforestation after 1996 were considered as the possible land source for reforestation due to the data limitation, but high rates of global mangrove area loss in the 1980s and 1990s^{40,41} were reported, proving potentially more available area for mangrove reforestation. Therefore, greater climate mitigation benefits and lower economic costs would be expected for mangrove reforestation.

All these points have been added into the main text and methods in Line 270–295, 380–388, and 586–635.

Fig. S7 Mitigation cost of mangrove reforestation at a national level. a, Implementation cost per megagram of carbon dioxide equivalent mitigated (\$ Mg CO₂-eq⁻¹). **b,** Total implementation and opportunity cost per megagram of carbon dioxide equivalent mitigated (\$ Mg CO₂-eq⁻¹). All the costs were adjusted into 2021 U.S. dollars.

Fig. 4 Carbon sequestration potential if mangrove afforestation and reforestation projects are implemented globally over a 40-year period. a-d, Changes in carbon density among pools over time during mangrove reforestation and afforestation. **a,** aboveground biomass carbon (AGC) pool (n=167 for reforestation, n=140 for afforestation); **b,** belowground biomass carbon (BGC) pool (n=129 for reforestation, n=124 for afforestation); **c,** sediment carbon (SCS) pool (n=51 for reforestation, n=108 for afforestation); and **d,** total ecosystem carbon (TECS) pool (n=29 for reforestation, n=63 for afforestation). Fitted lines and ribbons are calculated as the predicted values and the 95% confidence intervals for the 2.5th and 97.5th percentiles based on non-linear models. **e,** Country-specific maximum carbon mitigation potential from mangrove reforestation over 40 years (2021–2060) under the ‘1 year-completed restoration’ scenario at a cost threshold of \$100 Mg CO₂-eq⁻¹, with density of shading representing maximum carbon mitigation potential of each country. Countries in grey had no mangrove loss to deforestation between 1996 and 2016, or no mangroves. The inset graph represents the global maximum carbon mitigation potential by reforestation and assuming the same area for redeeming afforestation. Error bars represent 95% confidence interval for maximum carbon mitigation potential in each carbon pool.

Table S3. Global cumulative carbon sequestration potential during 2021-2060 (Tg CO₂-eq) of reforesting global feasible area and afforesting the same area under different scenarios. Scenario 1-4 assume different restoration rates (Methods).

Cost threshold		Scenario 1: 1 year-completed restoration	Scenario 2: 5 year-averaged restoration	Scenario 3: 10 year-averaged restoration	Scenario 4: varying rates across countries
\$10 Mg CO ₂ - eq ⁻¹	reforestation	251.5 (228.3, 274.6)	250.8 (229.4, 272.1)	249.4 (231.1, 267.8)	244.7 (232.7, 256.7)
	afforestation	158.4 (73.8, 242.9)	155.8 (80.5, 231.1)	152.0 (88.0, 215.9)	144.0 (95.0, 193.0)
\$50 Mg CO ₂ - eq ⁻¹	reforestation	630.2 (572.1, 688.3)	628.5 (575.0, 682.0)	625.1 (579.1, 671.1)	611.7 (583.9, 639.6)
	afforestation	396.9 (184.9, 608.7)	390.4 (201.7, 579.2)	380.9 (220.7, 541)	358.2 (240.5, 475.9)
\$100 Mg CO ₂ - eq ⁻¹	reforestation	630.8 (572.7, 688.9)	629.1 (575.5, 682.7)	625.7 (579.7, 671.8)	612.3 (584.4, 640.2)
	afforestation	397.3 (185.1, 609.3)	390.8 (201.9, 579.8)	381.2 (220.9, 541.6)	358.5 (240.7, 476.3)

Table S4. Reforestation and afforestation cost in developing and developed countries. All the cost were adjusted to 2021 U.S. dollars. Restoration implementation cost and opportunity cost of some countries that are not provided by Jakovac, et al. ³⁷ is adjusted from the average value into each country level based on their gross domestic product per capita (94 developing countries and 10 developed countries) (equation (6)).

		Developing countries				Developed countries				Reference
		mean	median	standard error	n	mean	median	standard error	n	
Reforestation	Implementation cost (\$)	13,410.6	1,122.4	10,178.9	17	130,345.2	47,619.8	32,615.7	39	³⁵
	Opportunity cost (\$)	4,295.3	4,191.2	208.8	83	7,029.3	5,720.2	1,276.2	9	³⁷
	Total cost (\$)	17,894.3	12,533.7	2,118.7	94	137,662.7	126,401.6	8,405.7	10	
	Unit cost (\$ Mg CO ₂ -eq ⁻¹)	16.0	11.2	1.9	94	123.1	113.0	7.5	10	
Afforestation	Implementation cost (\$)	16,457.3	2,410.6	9,571.8	8	155,843.3	60,532.8	96,176.1	8	³⁵
	Unit cost (\$ Mg CO ₂ -eq ⁻¹)	23.5	12.4	3.5	94	222.9	206.9	13.8	10	

Line 37: replace “same” with “the same”

Re: Revised accordingly.

Line 45-47: revise sentence structure “covering, e.g.,”.

Re: Thank you for your suggestion. We have changed the sentence into “Since the 1970s, mangrove restoration projects have been initiated in regions such as Southeast Asia, East Asia and South America^{2,3}.” (Line 43–44).

Lines 139-142: The claim “These data even..restoration sites” requires more justification and explanation for an audience as general at Nature Communications. Given the heavy framing around carbon storage, please provide additional context and framing around the related topic of wood production before making this leap, which is currently assumed to be understood by readers.

Re: Thank you for this suggestion. We added a figure to clarify this point and revised the sentences in Line 149–156:

The aboveground biomass carbon accumulation rate first increases with age, peaks around 10–15 years, and then decreases (Supplementary Fig. S3). Therefore, an effective rotation age for maximizing continuous wood production benefit (i.e., maximum wood production per year) might be around 10–15 years, but certainly less than 40 years. However, the logging practices would return some carbon to the atmosphere through residual wood decay and pulsed lateral carbon fluxes, and thus reduce the net carbon benefit in terms of climate mitigation.

Fig. S3 Average aboveground biomass carbon accumulation rate during mangrove reforestation for different age groups. Average aboveground biomass carbon accumulation rate is calculated by the biomass carbon density at each age divided by the age. The x-axis represents different age groups (0–5, 5–10, 10–15, 15–20, and 20–40 years). For each box plot, individual data points are shown as circles, the center line and the top and bottom of the box represent the median and the interquartile range (25th and 75th percentile). The whiskers represent the minimum and maximum limits. Outliers are represented by dots. Mean carbon density in each age group is represented by a circle and connected by a line. Sample size (n) of each group is shown at the bottom of each boxplot. Significance level of difference among age groups is calculated by the Kruskal–Wallis test combined with the Bonferroni adjusted post hoc Dunn test. Groups sharing the same letter are not significantly different at level of 0.05.

I must defer to my colleagues and other reviewers in the natural and physical sciences to assess the findings presented in Fig:2-3.

Re: We followed the suggestions of the other two reviewers and did some new analyses. These new results also support our main findings in Fig 2 and 3.

Lines 255-257: This headline number is impressive and worth presenting. However, it is idealistic to think that such comprehensive mangrove reforestation activities are feasible. In the same way that the authors exclude eroded and developed locations, it would make sense to conduct a more nuanced and pragmatic exclusion of areas that have such a higher opportunity cost of reforestation that they are irrelevant. Of course, many mangrove areas that were recently deforested experienced extraordinary pressure from land use change and reversing that pressure, in many cases, is unlikely. In other words, even though afforestation activities provide less carbon storage potential... they might be easier and more viable targets relative to areas that are socially prohibitive to reforest. Can you also present a refined number that focuses on areas with proposed or existing mangrove reforestation efforts and/or high likely and/or social and economic suitability for reforestation?

Re: Thank you for your suggestion. Following the reviewer's suggestion, we incorporated the economic constraints to exclude regions with high restoration costs and assume the remaining areas as the physically and economically suitable areas for reforestation. Please see the details in the first reply above.

Accordingly, we revised this sentence in Line 298–302 with a refined number after considering the economic constraints:

In terms of annual ecosystem productivity, restoring all potentially available mangrove areas (reforestation) would increase annual CO₂ uptake of existing estuarine and coastal wetland ecosystems by 1.7–4.3%⁴², highlighting the potential blue carbon sink strength of mangrove reforestation especially in regions suffering high levels of recent deforestation.

Lines 262-266: The same argument could be made for reforesting mangroves that will replace built and public services that arose from those deforestation activities with ecosystem services such as carbon storage. This is why the sub-analysis suggested in my previous comment is important as not to focus on the idealistic case of full reforestation without recognizing that many of these lands have high local post-

deforestation value that would render their reforestation cost prohibitive. Note the distinction in lines 266-268 “Furthermore, ... new mangrove habitat” the authors are referring to the cost of actual reforestation activities rather than the opportunity costs of land conversion that will be experienced by local communities that drove initially the deforestation of these mangroves. Therefore, it is important to present evidence specifically for areas that are primed to engage in such reforestation activities.

Re: Following the reviewer’s suggestion, we added new analyses by considering the implementation cost and opportunity cost for mangrove restoration. Please see the details in the first reply above.

Our new analysis indicates that even considering both direct implementation cost and opportunity cost, it is more cost-effective to rehabilitate or reforest existing mangrove areas ($\$16.0 \pm 1.9 \text{ Mg CO}_2\text{-eq}^{-1}$ in developing countries and $\$123.1 \pm 7.5 \text{ Mg CO}_2\text{-eq}^{-1}$ in developing countries) than to convert or create new mangrove habitat ($\$23.5 \pm 3.5 \text{ Mg CO}_2\text{-eq}^{-1}$ in developing countries and $\$222.9 \pm 13.8 \text{ Mg CO}_2\text{-eq}^{-1}$ in developing countries) (Supplementary Table S4, Line 309–314).

We believe these new analyses suggested by the reviewer made our manuscript much stronger and would like to thank again for the invested effort.

Reference

- 1 Soper, F. M. *et al.* Non-native mangroves support carbon storage, sediment carbon burial, and accretion of coastal ecosystems. *Global Change Biology* **25**, 4315-4326 (2019).
- 2 Duarte, C. M. *et al.* Rebuilding marine life. *Nature* **580**, 39-51 (2020).
- 3 Lee, S. Y., Hamilton, S., Barbier, E. B., Primavera, J. & Lewis, R. R. Better restoration policies are needed to conserve mangrove ecosystems. *Nature Ecology & Evolution* **3**, 870-872 (2019).
- 4 Hong, S. *et al.* Divergent responses of soil organic carbon to afforestation. *Nature Sustainability* **3**, 694-700 (2020).
- 5 Estrada-Villegas, S. *et al.* Edaphic factors and initial conditions influence

- successional trajectories of early regenerating tropical dry forests. *Journal of Ecology* **108**, 160-174 (2020).
- 6 Primavera, J. H., Rollon, R. N. & Samson, M. S. Mangrove rehabilitation in the Philippines: the challenges of coastal protection and pond reversion in *Treatise on Estuarine and Coastal Science* 217-244 (Waltham Academic Press, 2011).
 - 7 Wodehouse, D. C. J. & Rayment, M. B. Mangrove area and propagule number planting targets produce sub-optimal rehabilitation and afforestation outcomes. *Estuarine, Coastal and Shelf Science* **222**, 91-102 (2019).
 - 8 Krauss, K. W. *et al.* How mangrove forests adjust to rising sea level. *New Phytologist* **202**, 19-34 (2014).
 - 9 Agduma, A. R., Jiang, X., Liang, D. M., Chen, X. Y. & Cao, K. F. Stem hydraulic traits are decoupled from leaf ecophysiological traits in mangroves in southern Philippines. *Journal of Plant Biology* **65**, 389-401 (2022).
 - 10 Hilmi, E. *et al.* Tannins in mangrove plants in segara anakan lagoon, central java, indonesia. *Biodiversitas Journal of Biological Diversity* **22**, 3508-3516 (2021).
 - 11 Nordhaus, I., Salewski, T. & Jennerjahn, T. C. Interspecific variations in mangrove leaf litter decomposition are related to labile nitrogenous compounds. *Estuarine, Coastal and Shelf Science* **192**, 137-148 (2017).
 - 12 Mfilinge, P. L., Meziane, T., Bachok, Z. & Tsuchiya, M. Total lipid and fatty acid classes in decomposing mangrove leaves of *Bruguiera gymnorrhiza* and *Kandelia candel*: Significance with respect to lipid input. *Journal of Oceanography* **61**, 613-622 (2005).
 - 13 Kristensen, E., Bouillon, S., Dittmar, T. & Marchand, C. Organic carbon dynamics in mangrove ecosystems: A review. *Aquatic Botany* **89**, 201-219 (2008).
 - 14 Chanda, A. *et al.* Mangrove associates versus true mangroves: a comparative analysis of leaf litter decomposition in Sundarban. *Wetlands Ecology and Management* **24**, 293-315 (2016).
 - 15 Cameron, C., Hutley, L. B., Friess, D. A. & Brown, B. Community structure dynamics and carbon stock change of rehabilitated mangrove forests in Sulawesi, Indonesia. *Ecological Applications* **29**, e01810 (2019).

- 16 Kusumaningtyas, M. A. *et al.* Carbon sequestration potential in the rehabilitated mangroves in Indonesia. *Ecological Research* **37**, 80-91 (2022).
- 17 Djamaluddin, R., Brown, B. & Iii, R. The practice of hydrological restoration to rehabilitate abandoned shrimp ponds in Bunaken National Park, North Sulawesi, Indonesia. *Biodiversitas* **20**, 160-170 (2019).
- 18 Ola, A., Staples, T. L., Robinson, N. & Lovelock, C. E. Plasticity in the above- and below-ground development of mangrove seedlings in response to variation in soil bulk density. *Estuaries and Coasts* **43**, 111-119 (2020).
- 19 Oh, R. R. Y., Friess, D. A. & Brown, B. M. The role of surface elevation in the rehabilitation of abandoned aquaculture ponds to mangrove forests, Sulawesi, Indonesia. *Ecological Engineering* **100**, 325-334 (2017).
- 20 McKee, K. L., Cahoon, D. R. & Feller, I. C. Caribbean mangroves adjust to rising sea level through biotic controls on change in soil elevation. *Global Ecology and Biogeography* **16**, 545-556 (2007).
- 21 Adame, M. F., Cherian, S., Reef, R. & Stewart-Koster, B. Mangrove root biomass and the uncertainty of belowground carbon estimations. *Forest Ecology and Management* **403**, 52-60 (2017).
- 22 Castañeda-Moya, E. *et al.* Patterns of root dynamics in mangrove forests along environmental gradients in the Florida Coastal Everglades, USA. *Ecosystems* **14**, 1178-1195 (2011).
- 23 Kauffman, J. B. & Donato, D. C. *Protocols for the measurement, monitoring and reporting of structure, biomass and carbon stocks in mangrove forests.* (Center for International Forestry Research (CIFOR), 2012).
- 24 Howard, J., Hoyt, S., Isensee, K., Telszewski, M. & Pidgeon, E. *Coastal blue carbon: methods for assessing carbon stocks and emissions factors in mangroves, tidal salt marshes, and seagrasses.* (International Union for Conservation of Nature (IUCN), 2014).
- 25 Perdomo Trujillo, L. V., Mancera-Pineda, J. E., Medina-Calderon, J. H., Zimmer, M. & Schnetter, M. L. Massive loss of aboveground biomass and its effect on sediment organic carbon concentration: Less mangrove, more carbon? *Estuarine*,

- Coastal and Shelf Science* **248**, 106888 (2021).
- 26 Pham, V. H., Luu, V. D., Nguyen, T. T. & Koji, O. Will restored mangrove forests enhance sediment organic carbon and ecosystem carbon storage? *Regional Studies in Marine Science* **14**, 43-52 (2017).
- 27 He, Z. *et al.* Colonization by native species enhances the carbon storage capacity of exotic mangrove monocultures. *Carbon Balance and Management* **15**, 1-11 (2020).
- 28 Cameron, C. *et al.* Impact of an extreme monsoon on CO₂ and CH₄ fluxes from mangrove soils of the Ayeyarwady Delta, Myanmar. *Science of the Total Environment* **760**, 143422 (2021).
- 29 Murdiyarso, D., Sasmito, S. D., Sillanpää, M., MacKenzie, R. & Gaveau, D. Mangrove selective logging sustains biomass carbon recovery, soil carbon, and sediment. *Scientific Reports* **11**, 12325 (2021).
- 30 Nyland, R. D. Concepts of regeneration in *Silviculture: concepts and applications* Ch. 4, (Waveland Press, 2016).
- 31 Grassi, G. *et al.* Methods for estimation, measurement, monitoring and reporting of LULUCF activities under articles 3.3 and 3.4 in *2013 Revised supplementary methods and good practice guidance arising from the Kyoto protocol* (Intergovernmental Panel on Climate Change, 2014).
- 32 Macreadie, P. I. *et al.* Blue carbon as a natural climate solution. *Nature Reviews Earth & Environment* **2**, 826-839 (2021).
- 33 Griscom, B. W. *et al.* Natural climate solutions. *Proceedings of the National Academy of Sciences* **114**, 11645-11650 (2017).
- 34 Lu, N. *et al.* Biophysical and economic constraints on China's natural climate solutions. *Nature Climate Change* **12**, 847–853 (2022).
- 35 Bayraktarov, E. *et al.* The cost and feasibility of marine coastal restoration. *Ecological Applications* **26**, 1055-1074 (2016).
- 36 World Bank. *GDP per capita (current US\$)*. <https://data.worldbank.org/indicator/NY.GDP.PCAP.CD> (2021).
- 37 Jakovac, C. C. *et al.* Costs and carbon benefits of mangrove conservation and restoration: a global analysis. *Ecological Economics* **176**, 106758 (2020).

- 38 World Bank. *State and Trends of Carbon Pricing 2021*. (World Bank,, Washington, DC, 2021).
- 39 Zeng, Y. *et al.* Economic and social constraints on reforestation for climate mitigation in Southeast Asia. *Nature Climate Change* **10**, 842-844 (2020).
- 40 Wang, W. *et al.* *Report on China mangrove conservation and restoration strategy research project*. (Wetland Management Department, National Forestry and Grassland Bureau, 2020).
- 41 Gusmawati, N. *et al.* Surveying shrimp aquaculture pond activity using multitemporal VHSR satellite images - case study from the Perancak estuary, Bali, Indonesia. *Marine Pollution Bulletin* **131**, 49-60 (2018).
- 42 Regnier, P., Resplandy, L., Najjar, R. G. & Ciais, P. The land-to-ocean loops of the global carbon cycle. *Nature* **603**, 401-410 (2022).

Reviewers' Comments:

Reviewer #1:

Remarks to the Author:

The manuscript is clearly revised according to my comments and suggestions for the previous version. However, I would like to point out some minor typing errors. For example, the scientific name with "spp.", "spp." should not be in italic.

Reviewer #2:

Remarks to the Author:

The authors have adequately responded to my comments.

Reviewer #3:

Remarks to the Author:

The authors took extraordinary efforts to respond to my comments. Unfortunately, I was only asking the reviewers to recognize the limitations of their analysis with regard to the feasibility and viability of mangrove restoration projects globally. Still, in my view, a "benefits oriented" contribution and cataloguing of potential restoration is a novel contribution. The authors go down an excessive "rabbit hole" in attempting to track the opportunity cost of mangrove restoration activity, which is probably not necessary and I would have thought infeasible. However, the authors do a decent job with this analysis... but there are major new concerns. Firstly, the description and explanation of this analysis is highly unclear and difficult to follow compared to the initial manuscript.

Now that the authors have entered the social "cost-benefit" analysis world... rather than tracking "potential benefit flows", there seem to be some major violations of basic economic principles and standards. For example, I see that 40-years was picked as an arbitrary planning horizon, which is fine and consistent with the Ecological Economics article, but there is no mention of discounting benefit flows over time. Does this carry over from the Jakovac article where discounting did happen?

Also, the following sentence makes me concerned and confused:

"To evaluate the economic constraints on reforestation, we assumed that the mitigation cost should be lower than the three commonly used cost thresholds (i.e., \$10, \$50 and \$100 Mg CO₂-eq-1), otherwise it would be economically infeasible."

I'm confused about how the word "mitigation" is being used and why any such threshold would ever be assumed to ensure a program remains economically feasible. The initial critique levied in my first review was that it is unlikely that all potential restoration projects would be feasible, which was a shortcoming of the analysis as it relates to policy formation. Assuming away such constraints, if that is what the authors are doing, does not help the cause.

The authors should clarify this new analysis with the help of an economist who can communicate its limitations, relevance and technical rigor to a broad audience or remove the analysis in its entirety and recognize the shortcomings that I noted the first time around.

Replies to Reviewer's comment

Reviewer #1 (Remarks to the Author):

The manuscript is clearly revised according to my comments and suggestions for the previous version. However, I would like to point out some minor typing errors. For example, the scientific name with "spp.", "spp." should not be in italic.

Reply: Thank you very much for your positive comments on our manuscript and for pointing these typing errors. We changed all 'spp.' to be non-italic format (Line 233, 235, 236, 353, and 463).

Reviewer #2 (Remarks to the Author):

The authors have adequately responded to my comments.

Reply: Thank you very much for your positive assessment of our revision.

Reviewer #3 (Remarks to the Author):

The authors took extraordinary efforts to respond to my comments. Unfortunately, I was only asking the reviewers to recognize the limitations of their analysis with regard to the feasibility and viability of mangrove restoration projects globally. Still, in my view, a "benefits oriented" contribution and cataloguing of potential restoration is a novel contribution. The authors go down an excessive "rabbit hole" in attempting to track the opportunity cost of mangrove restoration activity, which is probably not necessary and I would have thought infeasible. However, the authors do a decent job with this analysis... but there are major new concerns. Firstly, the description and explanation of this analysis is highly unclear and difficult to follow compared to the initial manuscript.

Now that the authors have entered the social "cost-benefit" analysis world... rather than tracking "potential benefit flows", there seem to be some major violations of basic economic principles and standards. For example, I see that 40-years was picked as an arbitrary planning horizon, which is fine and consistent with the Ecological Economics

article, but there is no mention of discounting benefit flows over time. Does this carry over from the Jakovac article where discounting did happen?

Also, the following sentence makes me concerned and confused:

"To evaluate the economic constraints on reforestation, we assumed that the mitigation cost should be lower than the three commonly used cost thresholds (i.e., \$10, \$50 and \$100 Mg CO₂-eq-1), otherwise it would be economically infeasible."

I'm confused about how the word "mitigation" is being used and why any such threshold would ever be assumed to ensure a program remains economically feasible. The initial critique levied in my first review was that it is unlikely that all potential restoration projects would be feasible, which was a shortcoming of the analysis as it relates to policy formation. Assuming away such constraints, if that is what the authors are doing, does not help the cause.

The authors should clarify this new analysis with the help of an economist who can communicate its limitations, relevance and technical rigor to a broad audience or remove the analysis in its entirety and recognize the shortcomings that I noted the first time around.

Reply: Thank you very much for your comments and suggestions. We indeed misunderstood your suggestions during the previous round revision. In fact, we agreed with you on that a "benefits oriented" contribution and cataloguing of potential restoration is a novel contribution and that considering biophysical limitations of mangrove restoration is only the first step to assess the feasible region for future restoration projects. As you and the editor suggested, we now removed the additional analysis with regard to the feasibility and viability of mangrove restoration projects globally. Instead, we added some discussion on these limitations of our analyses (Line 365-384).